# Neural Common Neighbor with Completion for Link Prediction

**Xiyuan Wang**[1,2]
wangxiyuan@pku.edu.cn

**Haotong Yang**[1,2,3]
haotongyang@pku.edu.cn

**Muhan Zhang**[1*]
muhan@pku.edu.cn

[1]**Institute for Artificial Intelligence, Peking University.**
[2]School of Intelligence Science and Technology, Peking University.
[3]Key Lab of Machine Perception (MoE) *

## Abstract

In this work, we propose a novel link prediction model and further boost it by studying graph incompleteness. First, we introduce MPNN-then-SF, an innovative architecture leveraging structural feature (SF) to guide MPNN's representation pooling, with its implementation, namely Neural Common Neighbor (NCN). NCN exhibits superior expressiveness and scalability compared with existing models, which can be classified into two categories: SF-then-MPNN, augmenting MPNN's input with SF, and SF-and-MPNN, decoupling SF and MPNN. Second, we investigate the impact of graph incompleteness—the phenomenon that some links are unobserved in the input graph—on SF, like the common neighbor. Through dataset visualization, we observe that incompleteness reduces common neighbors and induces distribution shifts, significantly affecting model performance. To address this issue, we propose to use a link prediction model to complete the common neighbor structure. Combining this method with NCN, we propose Neural Common Neighbor with Completion (NCNC). NCN and NCNC outperform recent strong baselines by large margins, and NCNC further surpasses state-of-the-art models in standard link prediction benchmarks. Our code is available at https://github.com/GraphPKU/NeuralCommonNeighbor.

## 1 Introduction

Link prediction is a crucial task in graph machine learning, finding applications in various domains, such as recommender systems (Zhang & Chen, 2020), knowledge graph completion (Zhu et al., 2021), and drug interaction prediction (Souri et al., 2022). Graph Neural Networks (GNNs) have gained prominence in link prediction tasks, with Graph Autoencoder (GAE) (Kipf & Welling, 2016) being a notable representation. GAE utilizes Message Passing Neural Network (MPNN) (Gilmer et al., 2017) representations of two individual target nodes to predict link existence. However, Zhang et al. (2021) point out a limitation in GAE: it overlooks pairwise relations between target nodes. For example, in Figure 1, GAE always produces the same prediction for two links $(v_1, v_2)$ and $(v_1, v_3)$ despite their differing pairwise relationships, because MPNN generates identical representations for nodes $v_2, v_3$ due to graph symmetry. Nevertheless, the two links have different structural features. For example, $v_1$ and $v_2$ have a common neighbor $v_4$,

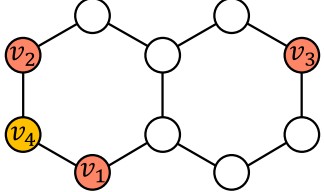

Figure 1: The failure of MPNN in link prediction task. $v_2$ and $v_3$ have equal MPNN node representations due to symmetry. However, with different pairwise relations, $(v_1, v_2)$ and $(v_1, v_3)$ should have different representations.

while $v_1$ and $v_3$ do not have any. Therefore, various methods combine structural feature (SF) and MPNN for better expressivity and have dominated the link prediction task (Zhang et al., 2021; Yun et al., 2021; Chamberlain et al., 2023).

---

*Correspondence to Muhan Zhang.

However, models combining SF and MPNN still have much room for improvement. They can generally be concluded into two architectures: *SF-then-MPNN* and *SF-and-MPNN*, as shown in Figure 2. SEAL (Zhang & Chen, 2018) adds target-link-specific hand-crafted features to the node features of the input graphs of MPNN, whose output node representations are then pooled to produce link representations. SEAL belongs to the SF-then-MPNN architecture, which leverages SF to augment the input graph of MPNN. Though SF-then-MPNN models achieve provably high expressivity (Zhang et al., 2021), they require running MPNN on a different graph for each target link, resulting in significant computational overhead.

In contrast, Neo-GNN (Yun et al., 2021) and BUDDY (Chamberlain et al., 2023) decouple the structural feature from MPNN. They directly incorporate manually created pairwise features with individual node representations produced by MPNN as link representations, necessitating only a single run of MPNN on the original graph. These models fall under the SF-and-MPNN category, where structural features and MPNN are independent. Such methods have limited expressivity. For example, SEAL can capture the representations of common neighbor nodes, while BUDDY can only count the number of common neighbors.

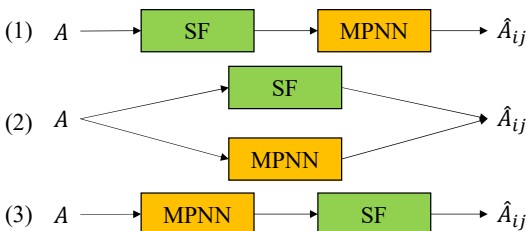

Figure 2: Archtectures for combining SF and MPNN. $A$ denote the input graph structure. Existing works are (1) SF-then-MPNN (2) SF-and-MPNN architectures. We propose a completely new architecture (3) MPNN-then-SF.

To solve the drawback of the two architectures above, we propose *MPNN-then-SF* architecture, which initially applies MPNN to the original graph and then uses structural features to guide the pooling of node representations. This approach offers strong expressivity and scalability: similar to SF-then-MPNN models, MPNN-then-SF can capture the node features of common neighbors, and similar to SF-and-MPNN models, it runs MPNN only once for all target links. We introduce the Neural Common Neighbor (NCN) as an instantiation of the MPNN-then-SF architecture. In experiments, NCN outperforms existing models in both scalability and performance.

Furthermore, since NCN heavily relies on common neighbor structure, which is significantly affected by graph incompleteness, we also investigate the impact of incompleteness. Graph incompleteness is ubiquitous in link prediction tasks because the goal is to predict unobserved edges not present in the input graph. We empirically observe that incompleteness reduces the number of common neighbor and leads to a shift in the distribution of common neighbor between the training and test sets. These phenomena collectively lead to performance degradation. To mitigate this issue, we first employ NCN to complete the common neighbor structure and then apply NCN to the completed graph. In experiments, this method significantly improves the performance of NCN.

In conclusion, our contributions are as follows:

- We introduce the Neural Common Neighbor (NCN) for link prediction using the MPNN-then-SF architecture, demonstrating superior performance and scalability compared to existing models.
- We analyze the impact of graph incompleteness and propose Neural Common Neighbor with Completion (NCNC), which completes the input common neighbor structure and applies NCN to the completed graph. NCNC outperforms state-of-the-art models.

## 2 PRELIMINARY

We consider an undirected graph $\mathcal{G} = (V, E, A, X)$, where $V = \{1, 2, \ldots, n\}$ represents a set of $n$ nodes, $E \subseteq V \times V$ denotes the set of edges, $X \in \mathbb{R}^{n \times F}$ is a node feature matrix whose $v$-th row $X_v$ is the feature of node $v$, and adjacency matrix $A \in \mathbb{R}^{n \times n}$ is a symmetric matrix, whose $(u, v)$ element is 1 if $(u, v) \in E$ and 0 otherwise. The *degree* of node $u$ is $d(u, A) := \sum_{v=1}^{n} A_{uv}$. Node $u$'s neighbors are nodes connected to $u$, $N(u, A) := \{v | v \in V, A_{uv} > 0\}$. For simplicity of notations, we use $N(u)$ to denote $N(u, A)$ when $A$ is fixed. *Common neighbor* means nodes connected to both $i$ and $j$: $N(i) \cap N(j)$.

**High Order Neighbor.** We define $A^l$ as a high-order adjacency matrix, where $A_{uv}^l$ represents the number of walks of length $l$ between nodes $u$ and $v$ in graph $A$. $N(u, A^l) = \{v | v \in V, A_{uv}^l > 0\}$ denotes the set of nodes connected to $u$ by a walk of length $l$ in graph $A$, equivalent to the neighbors in higher-order adjacency. $N_l(u, A)$ denotes the set of nodes whose shortest path distance to $u$ in graph $A$ is $l$. Existing works define *high-order neighbors* as either $N(u, A^l)$ or $N_l(u, A)$. More generally, the neighborhood of $u$ can be expressed as $N_{l_1}(u, A^{l_2})$, returning all nodes with a shortest path distance of $l_1$ to $u$ in the high-order graph $A^{l_2}$. We use $N_{l_1}^{l_2}(u)$ to denote $N_{l_1}(u, A^{l_2})$ when $A$ is fixed. Given a target link $(i, j)$, their general neighborhood overlap is given by $N_{l_1}^{l_2}(i) \cap N_{l_1'}^{l_2'}(j)$, and their neighborhood difference is given by $N_{l_1}^{l_2}(i) - N_{l_1'}^{l_2'}(j)$.

**Message Passing Neural Network (MPNN).** Comprising message passing layers, MPNN (Gilmer et al., 2017) is a common GNN framework. The $k^{\text{th}}$ layer is as follows.

$$\boldsymbol{h}_v^{(k)} = U^{(k)}(\boldsymbol{h}_v^{(k-1)}, \text{AGG}(\{M^{(k)}(\boldsymbol{h}_v^{(k-1)}, \boldsymbol{h}_u^{(k-1)}) | u \in N(v)\})), \tag{1}$$

where $\boldsymbol{h}_v^{(k)}$ is the representation of node $v$ at the $k^{\text{th}}$ layer, $U^{(k)}, M^{(k)}$ are functions like multi-layer perceptron (MLP), and AGG denotes an aggregation function like sum or max. The initial node representation $\boldsymbol{h}_v^{(0)}$ is node feature $X_v$. In each message passing layer, information is aggregated from neighbors to update the node representation. The final node representations produced by MPNN are the output of the last message passing layer, denoted as $\text{MPNN}(v, A, X) = \boldsymbol{h}_v^{(K)}$.

## 3 RELATED WORK

### 3.1 LINK PREDICTION MODEL

There are three primary categories of link prediction models. *Node embedding methods* (Perozzi et al., 2014; Grover & Leskovec, 2016; Tang et al., 2015) map each node to an embedding vector and combine the embeddings of target nodes to predict link. *Link prediction heuristics* (Liben-Nowell & Kleinberg, 2003; Barabási & Albert, 1999; Zhou et al., 2009; Adamic & Adar, 2003) develop hand-crafted structural features. *GNNs* utilize Graph Neural Networks to predict link existence. Among these GNNs, Graph Autoencoder (GAE) (Kipf & Welling, 2016) uses the inner product of the MPNN representations of two target nodes, $\langle \text{MPNN}(i, A, X), \text{MPNN}(j, A, X) \rangle$, as the representations of link $(i, j)$. It uses MPNN only and thus fails to capture pairwise relations between nodes. In contrast, various GNNs combining MPNN and structural features (Zhang & Chen, 2018; Yun et al., 2021; Chamberlain et al., 2023) have achieved state-of-the-art performance. Take SEAL (Zhang & Chen, 2018) as an example. For a target link $(i, j)$, SEAL initially augments node feature $X$ by concatenating each node's shortest path distance to $(i, j)$ and extracts a $k$-hop subgraph from the whole graph, generating augmented node feature $X'$ and adjacency $A'$, respectively. Subsequently, SEAL applies MPNN to this subgraph and use the sum of node representations within it, $\sum_u \text{MPNN}(u, A', X')$, as the target link representation. Other models employ a distinct approach to incorporate structural features. Neo-GNN (Yun et al., 2021) and BUDDY (Chamberlain et al., 2023), for instance, directly apply MPNN to the original graph and concatenate structural features, such as the count of common neighbors, with the Hadamard product of target node MPNN representations, $\text{MPNN}(i, A, X) \odot \text{MPNN}(j, A, X) \| \text{structural features}$.

### 3.2 STRUCTURAL FEATURE

Structural features for link prediction vary among models but are generally based on neighborhood overlap. Notable heuristics, such as Common Neighbor (CN), Resource Allocation (RA), and Adamic Adar (AA), use first-order common neighbors to compute scores for a target link $(i, j)$:

$$\text{CN}(i, j) = \sum_{u \in N(i) \cap N(j)} 1, \quad \text{RA}(i, j) = \sum_{u \in N(i) \cap N(j)} \frac{1}{d(u)}, \quad \text{AA}(i, j) = \sum_{u \in N(i) \cap N(j)} \frac{1}{\log d(u)}. \tag{2}$$

Neo-GNN (Yun et al., 2021) and BUDDY (Chamberlain et al., 2023) extend these heuristics by utilizing higher-order neighbors. Neo-

Table 1: Summary of existing models using Equation (5). FO and HO denote first-order and high-order neighbors, respectively. $\cap$ and $-$ denote set intersection and difference, respectively.

| | Model | Neighbor | $\oplus$ | $g(x)$ | $f(x)$ |
|---|---|---|---|---|---|
| | CN | FO | $\cap$ | 1 | 1 |
| Heuristics | RA | FO | $\cap$ | 1 | $1/x$ |
| | AA | FO | $\cap$ | 1 | $1/\log(x)$ |

GNN computes features for high-order neighborhood overlap $N^{l_1}(i), N^{l_2}(j)$ as follows,

$$\sum_{u \in N_1^{l_1}(i) \cap N_1^{l_2}(j)} A_{iu}^{l_1} A_{ju}^{l_2} f(d(u)), \qquad (3)$$

where $f$ is a learnable function of node degree $d(u)$. BUDDY (Chamberlain et al., 2023) further utilize high-order neighborhood difference. It computes overlap features $\{a_{l_1,l_2}(i,j)|l_1, l_2 = 1, 2, ..., k\}$ and difference features $\{b_l(i,j), b_l(j,i)|l = 1, 2, ..., k\}$ as follows:

$$a_{l_1,l_2}(i,j) = \sum_{u \in N_{l_1}^1(i) \cap N_{l_2}^1(j)} 1, \quad b_l(i,j) = \sum_{u \in N_l^1(i) - \bigcup_{l'=1}^k N_{l'}^1(j)} 1. \qquad (4)$$

All these pairwise features can be summarized into the following framework.

$$\sum_{u \in N_{l_1}^{l_2}(i) \oplus N_{l_1'}^{l_2'}(j)} g(A_{iu}^{l_2}) g(A_{ju}^{l_2'}) f(d(u)), \qquad (5)$$

where $N_{l_1}^{l_2}(i)$ and $N_{l_1'}^{l_2'}(j)$ denote the general neighborhood of $i$ and $j$, $\oplus$ is a set operator like intersection or difference, and $f, g$ are node degree and high-order adjacency weight functions, respectively. Details on how this framework unify existing structure features are shown in Table 1.

### 3.3 INCOMPLETENESS OF GRAPH

The primary aim of the link prediction task is to forecast unobserved edges, inherently making the input graph incomplete. Nevertheless, graph incompleteness can significantly impact structural features, such as common neighbors, and models based on them. This issue has drawn attention in some prior works. Yang et al. (2022) examined how unobserved links could distort evaluation scores, with a specific focus on metrics and benchmark design, while our research concentrates on model design. Dong et al. (2022) explored the consequences of the presence of target links, whereas our emphasis lies in understanding how incompleteness affects common neighbor-based features. Outside the domain of link prediction, Zhao et al. (2023) and Zhao et al. (2021) add edges predicted by GAE to the input graph of GNNs. However, their primary objective was node classification tasks, aiming to enhance edges between nodes of the same class while diminishing others. In contrast, our research addresses distribution shifts and information loss stemming from graph incompleteness, offering unique completion methods and insights tailored for link prediction.

## 4 NEURAL COMMON NEIGHBOR

Structural features (SF), such as common neighbors, are commonly employed in link prediction models. Existing approaches combine SF with Message Passing Neural Networks (MPNN) in two manners (illustrated in Figure 2): SF-then-MPNN and SF-and-MPNN. However, these approaches exhibit limitations in terms of either scalability or expressivity. To address these issues comprehensively, we introduce a novel architecture, MPNN-then-SF, which offers a unique blend of high expressivity and scalability. Subsequently, we present a concrete instantiation of this architecture, Neural Common Neighbor (NCN). All proofs for theorems in this section are in Appendix A.

### 4.1 NEW ARCHITECTURE COMBINING MPNN AND SF

In Figure 2, we categorize existing methods into two architectures:

- SF-then-MPNN: This category includes SEAL (Zhang & Chen, 2018) and NBFNet (Zhu et al., 2021). In this approach, the input graph is initially enriched with structural features and then fed into the MPNN, which allows MPNN to leverage SF and have provable expressivity (Zhang & Chen, 2018). However, the drawback is that structural features change with target link, necessitating MPNN to be re-run for each link, resulting in lower scalability.

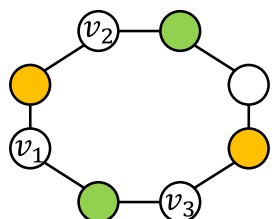

Figure 3: White, green, and yellow colors represent node features $0, 1,$ and $2$, respectively. Both links $(v_1, v_2)$ and $(v_1, v_3)$ have one common neighbor, making it indistinguishable for existing SF-and-MPNN models.

- SF-and-MPNN: This category encompasses models like NeoGNN (Yun et al., 2021) and BUDDY (Chamberlain et al., 2023). Here, MPNN takes the original graph as input and runs only once for all target links, leading to high scalability. However, SF are directly concatenated to the final representations and thus detached from MPNN, leading to reduced expressivity.

From these two architectural paradigms, it becomes apparent that feeding the original graph to MPNN is essential for achieving high scalability. Moreover, the coupling between SF and MPNN remains a crucial factor for expressivity. Thus, we introduce a new architecture: MPNN-then-SF. This approach initially runs MPNN on the original graph and then employs structural features to guide the pooling of MPNN features, requiring only one MPNN run and enhancing expressivity. The specific representation of the target link $(i, j)$ is as follows:

$$\text{Pool}(\{\text{MPNN}(u, A, X) | u \in S\}), \tag{6}$$

where Pool is a pooling function mapping a multiset of node representations to a single set representation, and $S$ is a node set related to the target link. Multiple node sets can be used in conjunction to produce concatenated representations. This flexible framework can express various models. When using target nodes $i$ and $j$ as $S$ and Hadamard product as Pool, it can express GAE:

$$\text{MPNN}(i, A, X) \odot \text{MPNN}(j, A, X). \tag{7}$$

Alternatively, we can choose $S$ as combinations of high-order neighbor sets of $i$ and $j$, leading to the following form (see Appendix A.1 for the detailed derivation.):

$$\sum_{u \in N_{l_1}^{l_2}(i) \oplus N_{l_1'}^{l_2'}(j)} g(A_{iu}^{l_2}) g(A_{ju}^{l_2'}) \text{MPNN}(u, A, X), \tag{8}$$

where $g$ is a function transforming the edge weight in high-order adjacency matrix. This framework exhibits stronger expressivity than existing SF-and-MPNN models.

**Theorem 1.** *Combination of Equation 7 and Equation 8 are strictly more expressive than MPNN-only model: GAE, SF-only models: CN, RA, AA, and MPNN-and-SF models: Neo-GNN, BUDDY.*

A key factor contributing to its higher expressivity is the coupling of MPNN and SF. While SF-and-MPNN typically only counts the number of common neighbors, MPNN-then-SF, similar to SF-then-MPNN, can capture node properties of these common neighbors. As shown in Figure 3, node tuples $(v_1, v_2)$ and $(v_1, v_3)$ have the same number of common neighbors. However, their common neighbors have different node features, allowing MPNN-then-SF and SF-then-MPNN to distinguish them, a capability that SF-and-MPNN lacks.

## 4.2 Neural Common Neighbor

We will now present an implementation for the MPNN-then-SF framework. Notably, the previous models NeoGNN (Yun et al., 2021) and BUDDY (Chamberlain et al., 2023) all incorporate higher-order neighbors into their architectures, resulting in significant performance improvements. Surprisingly, in our experiments, we observed that the gains achieved by explicitly considering higher-order neighbors were marginal once we introduced MPNN into the framework (as discussed in Section 6.3). We speculate that this marginal improvement arises because MPNN implicitly learns information related to higher-order neighbors. Therefore, considering scalability, we opt to utilize only the target nodes and their first-order common neighbors as the node set, leading to the development of our NCN model:

$$\text{NCN}(i, j, A, X) = \text{MPNN}(i, A, X) \odot \text{MPNN}(j, A, X) \Big\| \sum_{u \in N(i) \cap N(j)} \text{MPNN}(u, A, X) \tag{9}$$

where $g(A_{iu})$ and $g(A_{ju})$ are constants and ignored, and $\|$ denotes concatenation. It has high expressivity.

**Theorem 2.** *NCN is strictly more expressive than GAE, CN, RA, AA. Moreover, Neo-GNN and BUDDY are not more expressive than NCN.*

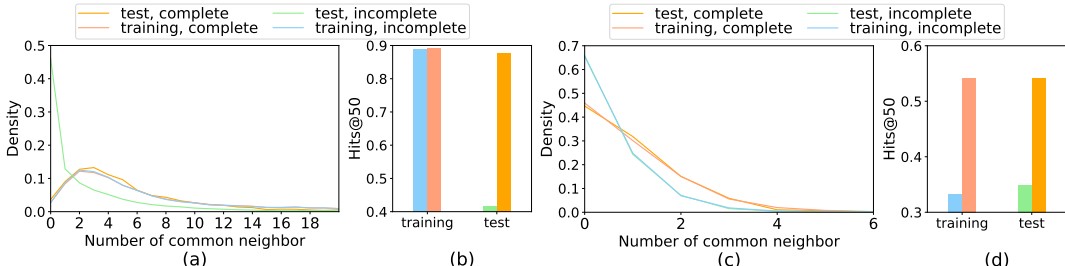

Figure 4: Visualization of incompleteness on datasets. The incomplete graph only contains edges in the training set, and the complete graph further contains edges in the validation and test set. (a) and (b) visualize the ogbl-collab dataset. (c) and (d) visualize the Cora dataset. (a) and (c) are for distributions of the number of common neighbors of the training edges and test edges. (b) and (d) show performance of CN on the training set and test set.

To elaborate, in certain scenarios where the properties of common neighbors hold significant importance, NCN outperforms both BUDDY and Neo-GNN in expressiveness.

As our first major contribution, NCN represents a straightforward yet potent model for combining structural features and MPNNs. It operates as an implicit high-order model by aggregating first-order common neighbors, each of which implicitly learns higher-order information through MPNN. A comprehensive analysis of time complexity is in Appendix E.

## 5 NEURAL COMMON NEIGHBOR WITH COMPLETION

While NCN outperforms existing models, it relies heavily on the common neighbor structure, which the incompleteness of the graph can notably influence. For instance, in cases where node pairs lack common neighbors, NCN essentially degenerates to GAE, rendering it unable to leverage structural features. Although the absence of common neighbors can suggest that a link is unlikely to exist, certain node pairs may possess common neighbors in the ground truth that remain unobserved in the input graph due to graph incompleteness. Graph incompleteness is ubiquitous in link prediction tasks, given that the objective is to predict unobserved edges. However, few studies have delved into this issue. In this section, we initially demonstrate that incompleteness can result in the loss of common neighbor information, distribution shifts between the training and test sets, and the consequent deterioration of model performance. We propose a straightforward yet effective method to tackle these challenges: common neighbor completion (CNC). CNC completes unobserved common neighbors using a link prediction model. With the introduction of CNC, we enhance NCN and introduce Neural Common Neighbor with Completion (NCNC).

### 5.1 INCOMPLETENESS VISUALIZATION

To illustrate the challenges posed by incompleteness, we analyze two common datasets: ogbl-collab (Hu et al., 2020) and Cora (Yang et al., 2016). We refer to the graph containing only the edges from the training set as the *incomplete* graph, while the one encompassing edges from the training, validation, and test sets is termed the *complete* graph.

Given the pivotal role of common neighbor information in our NCN model and other link prediction models, we visualize the distribution of the number of common neighbors for training/test edges in both complete and incomplete graphs separately in Figure 4 (a)(c). To assess how incompleteness impacts model performance, we present the performance of CN model (as shown in Section 3.2) in four distinct scenarios in Figure 4 (b) (d). We observe the following effects of incompleteness:

**Loss of Common Neighbors.** Figure 4(c) illustrates that in the incomplete graph, there are fewer common neighbors for both training and test sets, as indicated by the comparison between the blue (green) and red (orange) lines. When comparing the incomplete and complete graphs, it becomes evident that the incomplete graph suffers from a loss of common neighbor information due to incompleteness. Additionally, more links have no common neighbors at all.

**Common Neighbor Distribution Shift.** A noticeable *distribution shift* between the training and test sets is evident in the incomplete graph of the ogbl-collab dataset, as seen in the comparison

between the blue and green lines in Figure 4(a). This shift disappears when the graph is complete (the red and orange lines), indicating that incompleteness is the cause. Such a substantial distribution shift between training and test links could pose challenges in model generalization. This distribution shift is related to the dataset split method. Ogbl-collab is splitted based on the timestamp of edges, and the test edges all belong to the same year. Consequently, compared to the training edges, test edges exhibit stronger correlations with other test edges, resulting in a greater loss of common neighbor when these test edges are absent from the incomplete graph. Conversely, the Cora dataset is randomly splitted, so training and test edges lose a similar ratio of common neighbors and does not exhibit distribution shifts (Figure 4(c)).

**Performance Degradation.**    The performance of CN aligns with the common neighbor distribution. In the ogbl-collab dataset, the common neighbor distribution is nearly identical for the training set in both the complete and incomplete graphs, as is the performance (See Figure 4 (b)). However, test performance on the incomplete graph decreases significantly as the test distribution changes. Similar trends are observed in the Cora dataset, with test and training scores declining on incomplete graphs when the common neighbor distribution changes compared to the complete graph.

**Remark.**    Note that while common neighbor distribution changes may not fully account for the differences between complete and incomplete graphs, they offer valuable insights into how incompleteness alters the input graph structure for other learnable models. Despite CN is non-learnable and non-generalizable, its calculation for a target edge doesn't involve the edge itself, thereby avoiding data leakage concerns. These findings suggest that having a more complete input graph could yield superior link prediction models. However, in practice, we can only work with the incomplete input graph, necessitating exploring other mitigation methods for these issues.

## 5.2   COMMON NEIGHBOR COMPLETION

Motivated by the analysis above, we address graph incompleteness issues with a two-step method:

**Soft Completion of Common Neighbors.**    We start by softly completing the input graph with a link prediction model, such as NCN. However, instead of completing all edges in the entire graph, which can be impractical for large graphs, we focus specifically on common neighbor links. We compute the probability that a node $u$ serves as a common neighbor for a node tuple $(i, j)$ as follows:

$$P_{uij} = \begin{cases} 1 & \text{if } u \in N(i, A) \cap N(j, A) \\ \hat{A}_{iu} & \text{if } u \in N(j, A) - N(i, A) \\ \hat{A}_{ju} & \text{if } u \in N(i, A) - N(j, A) \\ 0 & \text{otherwise} \end{cases} \tag{10}$$

where $\hat{A}_{iu}$ represents the predicted existence probability of link $(i, u)$ by the model. The idea is that $u$ is a common neighbor of $(i, j)$ iff both edges $(i, u)$ and $(j, u)$ exist. If one of these edges is unobserved, we use NCN to predict its link existence probability, which we also use as the probability of $u$ being a common neighbor. In the rare case where both $(i, u)$ and $(j, u)$ are unobserved, we set the probability to 0. This technique is called "Common Neighbor Completion" (CNC).

**Reapplication of NCN on the Completed Graph.**    Following CNC, we apply the NCN model again on the graph that has been completed using the soft common neighbor weights $P_{uij}$. This final model is named **Neural Common Neighbor with Completion** (NCNC) and is defined as:

$$\text{NCNC}(i, j, A, X) = \text{MPNN}(i, A, X) \odot \text{MPNN}(j, A, X) \| \sum_{u \in N(i) \cup N(j)} P_{uij} \text{MPNN}(u, A, X). \tag{11}$$

Notably, the input graph of MPNN still takes the original graph as input, allowing it to run only once for all target links, thus maintaining high scalability. While $P_{uij}$ can be predicted using any link prediction model, weak models may not accurately recover the unobserved common neighbor structure. Therefore, in practice, we employ NCN to complete it.

In addition to addressing distribution shifts and common neighbor loss, NCNC also solves the problem that NCN can degrade to GAE when node pairs lack common neighbors. With NCNC, common

Table 2: Results on link prediction benchmarks. The format is average score ± standard deviation. OOM means out of GPU memory.

| | Cora | Citeseer | Pubmed | Collab | PPA | Citation2 | DDI |
|---|---|---|---|---|---|---|---|
| Metric | HR@100 | HR@100 | HR@100 | HR@50 | HR@100 | MRR | HR@20 |
| **CN** | $33.92_{\pm0.46}$ | $29.79_{\pm0.90}$ | $23.13_{\pm0.15}$ | $56.44_{\pm0.00}$ | $27.65_{\pm0.00}$ | $51.47_{\pm0.00}$ | $17.73_{\pm0.00}$ |
| **AA** | $39.85_{\pm1.34}$ | $35.19_{\pm1.33}$ | $27.38_{\pm0.11}$ | $64.35_{\pm0.00}$ | $32.45_{\pm0.00}$ | $51.89_{\pm0.00}$ | $18.61_{\pm0.00}$ |
| **RA** | $41.07_{\pm0.48}$ | $33.56_{\pm0.17}$ | $27.03_{\pm0.35}$ | $64.00_{\pm0.00}$ | $49.33_{\pm0.00}$ | $51.98_{\pm0.00}$ | $27.60_{\pm0.00}$ |
| **GCN** | $66.79_{\pm1.65}$ | $67.08_{\pm2.94}$ | $53.02_{\pm1.39}$ | $44.75_{\pm1.07}$ | $18.67_{\pm1.32}$ | $84.74_{\pm0.21}$ | $37.07_{\pm5.07}$ |
| **SAGE** | $55.02_{\pm4.03}$ | $57.01_{\pm3.74}$ | $39.66_{\pm0.72}$ | $48.10_{\pm0.81}$ | $16.55_{\pm2.40}$ | $82.60_{\pm0.36}$ | $53.90_{\pm4.74}$ |
| **SEAL** | $81.71_{\pm1.30}$ | $83.89_{\pm2.15}$ | $75.54_{\pm1.32}$ | $64.74_{\pm0.43}$ | $48.80_{\pm3.16}$ | $87.67_{\pm0.32}$ | $30.56_{\pm3.86}$ |
| **NBFnet** | $71.65_{\pm2.27}$ | $74.07_{\pm1.75}$ | $58.73_{\pm1.99}$ | OOM | OOM | OOM | $4.00_{\pm0.58}$ |
| **Neo-GNN** | $80.42_{\pm1.31}$ | $84.67_{\pm2.16}$ | $73.93_{\pm1.19}$ | $57.52_{\pm0.37}$ | $49.13_{\pm0.60}$ | $87.26_{\pm0.84}$ | $63.57_{\pm3.52}$ |
| **BUDDY** | $88.00_{\pm0.44}$ | $\underline{92.93_{\pm0.27}}$ | $74.10_{\pm0.78}$ | $\underline{65.94_{\pm0.58}}$ | $49.85_{\pm0.20}$ | $87.56_{\pm0.11}$ | $78.51_{\pm1.36}$ |
| **NCN** | $\underline{89.05_{\pm0.96}}$ | $91.56_{\pm1.43}$ | $\underline{79.05_{\pm1.16}}$ | $64.76_{\pm0.87}$ | $\underline{61.19_{\pm0.85}}$ | $88.09_{\pm0.06}$ | $82.32_{\pm6.10}$ |
| **NCNC** | $\mathbf{89.65_{\pm1.36}}$ | $\mathbf{93.47_{\pm0.95}}$ | $\mathbf{81.29_{\pm0.95}}$ | $\mathbf{66.61_{\pm0.71}}$ | $\mathbf{61.42_{\pm0.73}}$ | $\mathbf{89.12_{\pm0.40}}$ | $\mathbf{84.11_{\pm3.67}}$ |

Table 3: Ablation study on link prediction benchmarks.

| | Cora | Citeseer | Pubmed | Collab | PPA | Citation2 | DDI |
|---|---|---|---|---|---|---|---|
| Metric | HR@100 | HR@100 | HR@100 | HR@50 | HR@100 | MRR | HR@20 |
| **CN** | $33.92_{\pm0.46}$ | $29.79_{\pm0.90}$ | $23.13_{\pm0.15}$ | $56.44_{\pm0.00}$ | $27.65_{\pm0.00}$ | $51.47_{\pm0.00}$ | $17.73_{\pm0.00}$ |
| **GAE** | $89.01_{\pm1.32}$ | $91.78_{\pm0.94}$ | $78.81_{\pm1.64}$ | $36.96_{\pm0.95}$ | $19.49_{\pm0.75}$ | $79.95_{\pm0.09}$ | $61.53_{\pm9.59}$ |
| **GAE+CN** | $88.61_{\pm1.31}$ | $91.75_{\pm0.98}$ | $79.04_{\pm0.83}$ | $64.47_{\pm0.14}$ | $51.83_{\pm0.58}$ | $87.81_{\pm0.06}$ | $80.71_{\pm5.56}$ |
| **NCN2** | $88.87_{\pm1.34}$ | $91.36_{\pm1.02}$ | $80.21_{\pm0.78}$ | $65.43_{\pm0.46}$ | OOM | OOM | OOM |
| **NCN-diff** | $89.12_{\pm1.04}$ | $91.96_{\pm1.23}$ | $80.28_{\pm0.88}$ | $64.08_{\pm0.40}$ | $57.86_{\pm1.26}$ | $86.68_{\pm0.16}$ | $17.67_{\pm8.70}$ |
| **NCN** | $89.05_{\pm0.96}$ | $91.56_{\pm1.43}$ | $79.05_{\pm1.16}$ | $64.76_{\pm0.87}$ | $61.19_{\pm0.85}$ | $88.09_{\pm0.06}$ | $82.32_{\pm6.10}$ |
| **NCNC** | $89.65_{\pm1.36}$ | $93.47_{\pm0.95}$ | $81.29_{\pm0.95}$ | $66.61_{\pm0.71}$ | $61.42_{\pm0.73}$ | $89.12_{\pm0.40}$ | $84.11_{\pm3.67}$ |

neighbors are always completed, and the model only degenerates to GAE when both target nodes are isolated nodes. In such cases, where no structural features can be utilized, relying solely on the target node representations is reasonable. For a visual demonstration of CNC's effect, please refer to Appendix H, which illustrates how NCN can make more precise predictions by completing common neighbors for node pairs with no observed common neighbors.

# 6 EXPERIMENT

In this section, we extensively evaluate the performance of both NCN and NCNC. Detailed experimental settings are included in Appendix D.

We use seven popular real-world link prediction benchmarks. Among these, three are Planetoid citation networks: Cora, Citeseer, and Pubmed (Yang et al., 2016). Others are from Open Graph Benchmark (Hu et al., 2020): ogbl-collab, ogbl-ppa, ogbl-citation2, and ogbl-ddi. Their statistics and splits are shown in Appendix B.

## 6.1 EVALUATION ON REAL-WORLD DATASETS

In our evaluation on real-world datasets, we employ a range of baseline methods, encompassing traditional heuristics like CN (Barabási & Albert, 1999), RA (Zhou et al., 2009), and AA (Adamic & Adar, 2003), as well as GAE models, such as GCN (Kipf & Welling, 2017) and SAGE (Hamilton et al., 2017). Additionally, we consider SF-then-MPNN models, including SEAL (Zhang & Chen, 2018) and NBFNet (Zhu et al., 2021), as well as SF-and-MPNN models like Neo-GNN (Yun et al., 2021) and BUDDY (Chamberlain et al., 2023). The baseline results are sourced from (Chamberlain et al., 2023). Our models consist of NCN and NCNC. Their architectures are detailed in Appendix C.

The experimental results are presented in Table 2. NCN surpasses all baselines on 5/7 datasets and exhibits an average score improvement of 5% compared to BUDDY, the most competitive baseline.

Even on the remaining two datasets, NCN outperforms all baselines except BUDDY. These impressive results underscore the outstanding expressivity of our MPNN-then-SF architecture. Furthermore, NCNC enhances performance by an additional 2%, emerging as the top-performing method on all datasets. Notably, on ogbl-ppa, NCNC achieves an HR@100 score of 61.42%, surpassing the strongest baseline BUDDY by a substantial margin of over 10%. It's worth mentioning that our models outperform node embedding techniques (Perozzi et al., 2014; Grover & Leskovec, 2016; Tang et al., 2015) and other GNNs lacking pairwise features (Wang et al., 2021; 2022) significantly (see Appendix G).

## 6.2 SCALABILITY

We compare the inference time and GPU memory on ogbl-collab in Figure 5. NCN and NCNC have a **similar computation overhead to GAE**, as they both need to run MPNN only once. In contrast, SEAL, which reruns MPNN for each target link, takes 86 times more time compared with NCN with a small batch size 2048, and the disadvantage will be more significant with a larger batch size. Surprisingly, BUDDY and Neo-GNN are slower than NCN. The reason is that it uses pairwise features depending on high order neighbors that are much more time-consuming than common neighbor. NCN and NCNC also achieve low

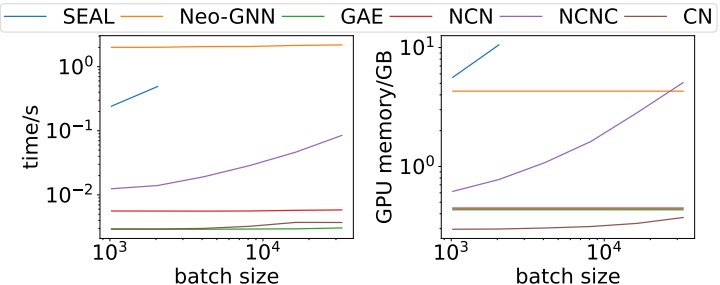

Figure 5: Inference time and GPU memory on ogbl-collab. The process we measure includes preprocessing and predicting one batch of test links. As shown in Appendix E, relation between time $y$ and batch size $t$ is $y = B + Ct$, where $B, C$ are model-specific constants. SEAL has out-of-memory problem and only uses small batch sizes.

GPU memory consumption. We also conduct scalability comparisons on other datasets and observe the same results (see Appendix F).

## 6.3 ABLATION ANALYSIS

To assess the effectiveness of the NCNC design, we conducted a comprehensive ablation analysis, as presented in Table 3.

Starting with GAE, which relies solely on node representations, we introduced GAE+CN, which incorporates Common Neighbor (CN) as pairwise features. Remarkably, GAE+CN outperforms GAE by 70% on Open Graph Benchmark (OGB) datasets, illustrating the importance of structural features. Furthermore, NCN exhibits a 5.5% score increase over GAE+CN, highlighting the advantages of the MPNN-then-SF architecture over the MPNN-and-SF architecture.

We also explore variants of NCN, namely NCN-diff and NCN2. In NCN-diff, we include neighborhood difference information by summing the representations of nodes in $N(i, A) - N(j, A)$ and $N(j, A) - N(i, A)$, while NCN2 incorporates high-order neighborhood overlap using $N(i, A^2) \cap N(j, A)$ and $N(i, A) \cap N(j, A^2)$. Notably, NCN, NCN-diff, and NCN2 exhibit similar performances across most datasets, suggesting that first-order neighborhood overlap might be sufficient. However, NCN-diff achieves a lower score on the DDI dataset, possibly because the high node degree in DDI introduces noisy and uninformative neighborhood difference information.

## 7 CONCLUSION

In this work, we introduce Neural Common Neighbor (NCN), a scalable and robust model for link prediction that harnesses the power of learnable pairwise features. Additionally, we address the challenge of graph incompleteness by identifying and visualizing common neighbor loss and distribution shifts stemming from this issue. To mitigate these problems, we introduce the Common Neighbor Completion (CNC) technique. Combining CNC with NCN, our final model, Neural Common

Neighbor with Completion (NCNC), outperforms state-of-the-art baselines across various datasets in terms of both speed and prediction performance.

## 8 LIMITATIONS

Though we propose MPNN-then-SF framework, we do not exhaust the design space and only propose one implementation, NCN, and its variants NCN2 and NCN-diff in ablation study. Moreover, while we only analyze the impact of incompleteness on common neighbor structures, graph incompleteness can also affect other structural features. Additionally, the proposed completion method has the potential to be generalized to address other structural features. Our future research will explore the design space of MPNN-then-SF and the broader implications of incompleteness on various structural features.

## 9 REPRODUCIBILITY STATEMENT

Our code is available at `https://github.com/GraphPKU/NeuralCommonNeighbor`. Proofs of all theorems in the maintext are in Appendix A.

## ACKNOWLEDGEMENT

This work is partially supported by the National Key R&D Program of China (2022ZD0160303), the National Natural Science Foundation of China (62276003), and Alibaba Innovative Research Program.

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

# A  PROOF

## A.1  DERIVATION OF EQUATION 8

The MPNN-then-SF architecture is as follows,

$$\text{Pool}(\{\!\!\{\text{MPNN}(u, A, X)|u \in S\}\!\!\}) \tag{12}$$

Let $S_{ab}$ be the following set,

$$(N_{l_1}^{l_2}(i) \oplus N_{l_1'}^{l_2'}(j)) \cap \{u \in V|A_{iu}^{l_2} = a\} \cap \{u \in V|A_{uj}^{l_2'} = b\}. \tag{13}$$

Then, for $S_{ab}$, we set the pooling function to sum and multiplied with $g(a)g(b)$, where $g$ is a function with high-order adjacency edge weight as input. Then the MPNN-then-SF architecture can express,

$$\sum_{u \in S_{ab}} g(a)g(b)\text{MPNN}(u, A, X) \tag{14}$$

Simply sums the feature of all $S_{ab}$ leads to,

$$\sum_{u \in N_{l_1}^{l_2}(i) \oplus N_{l_1'}^{l_2'}(j)} g(A_{iu}^{l_2})g(A_{ju}^{l_2'})\text{MPNN}(u, A, X). \tag{15}$$

## A.2  PROOF OF THEOREM 1 AND 2

Here, we present the theoretical proof of MPNN-then-SF's higher expressivity. We say algorithm A is strictly more expressive than algorithm B when A can differentiate all pairs of links that B can differentiate, while there exists a pair of links that A can distinguish while B cannot. We first prove the more expressive results by simulating other models with SF-then-MPNN and NCN then prove the strictness by constructing an example.

**Lemma 1.** *Equation 7 and NCN are more expressive than Graph Autoencoder (GAE)*

*Proof.* Graph Autoencoder's prediction for link (i, j) is $\langle\text{MPNN}(i, A, X), \text{MPNN}(j, A, X)\rangle$. So directly sum Equation 7 leads to GAE. Equation 7 is a part of NCN, so NCN can also express GAE. □

**Lemma 2.** *NCN is more expressive than CN,RA, and AA. Combination of Equation 7 and Equation 8 is more expressive than CN,RA,AA, BUDDY and Neo-GNN*

*Proof.* As MPNN can learn arbitrary functions of node degrees, NCN can express Equation 2, and Equation 8 can express the general form of structure feature 5. □

Furthermore, we construct an example in Figure 3. In that graph, $v_2$ and $v_3$ are symmetric and thus have the same MPNN representation, so GAE cannot distinguish $(v_1, v_2)$ and $(v_1, v_3)$. Moreover, $(v_1, v_2)$ and $(v_1, v_3)$ are symmetric if the node feature is ignored, so CN, RA, AA, Neo-GNN, BUDDY cannot distinguish them. However, $(v_1, v_2)$ have a common neighbor with feature 2, and $(v_1, v_3)$ have a common neighbor with feature 1, so NCN can distinguish them.

# B  DATASET STATISTICS

The statistics of each dataset are shown in Table 4.

Random splits use $70\%/10\%/20\%$ edges for training/validation/test set respectively. Different from others, the collab dataset allows using validation edges as input on test set.

# C  MODEL ARCHITECTURE

This section concludes our methods in Section 4 and Section 5.

Given an input graph $A$, a node feature matrix $X$, and target links $\{(i_1, j_1), (i_2, j_2), ..., (i_t, j_t)\}$, our models consist of three steps: target link removal, MPNN, and predictor. NCN and NCNC only differ in the last step. The model architecture is visualized in Figure 6

Table 4: Statistics of dataset.

| | **Cora** | **Citeseer** | **Pubmed** | **Collab** | **PPA** | **DDI** | **Citation2** |
|---|---|---|---|---|---|---|---|
| #Nodes | 2,708 | 3,327 | 18,717 | 235,868 | 576,289 | 4,267 | 2,927,963 |
| #Edges | 5,278 | 4,676 | 44,327 | 1,285,465 | 30,326,273 | 1,334,889 | 30,561,187 |
| splits | random | random | random | fixed | fixed | fixed | fixed |
| average degree | 3.9 | 2.74 | 4.5 | 5.45 | 52.62 | 312.84 | 10.44 |

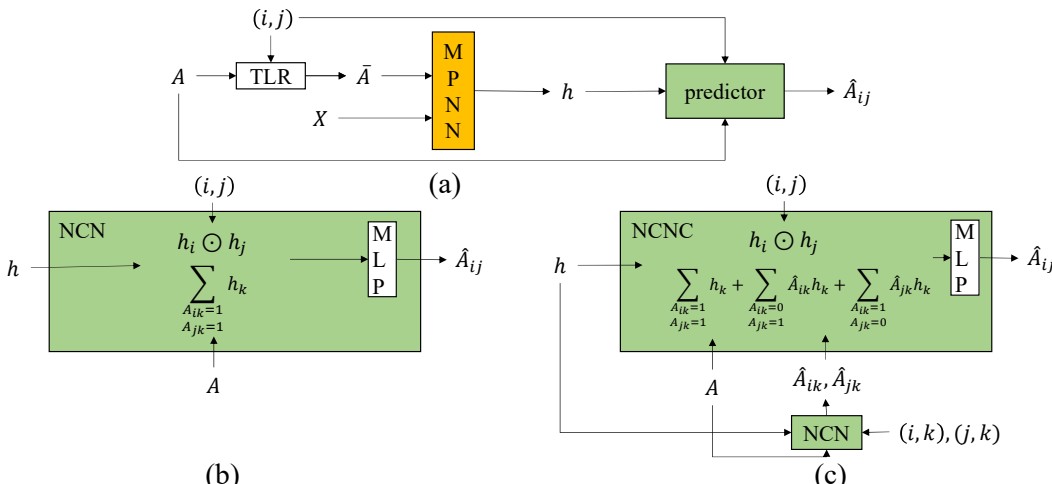

Figure 6: Architecture of our models. (a) The overall architecture. Given node feature $X$, adjacency matrix $A$ and target link $(i,j)$, models first set $A_{ij} = 0$ ( Target link removal, TLR). Then, $\bar{A}, X$ are fed to a vanilla MPNN for node representations $h$. With $(i,j)$, $h$, and $A$ as input, the predictor produces $\hat{A}_{ij}$, the probability that edge $(i,j)$ exists. (b) The NCN predictor. It uses node representations of target nodes $i, j$ and their common neighbors to produce edge representations. Then, it feeds the edge representation to an MLP to produce the final prediction. (c) The NCNC predictor. It first uses NCN to predict unobserved links $\hat{A}_{ik}, \hat{A}_{jk}$, which is then used to complete unobserved common neighbors.

**Target link removal.**  We make no changes to the input graph in the validation and test set where the target links are unobserved. In the training set, we remove target links from $A$. Let $\bar{A}$ denote the processed graph. This method is detailed in Section 5.

**MPNN.**  We use MPNN to produce node representations $h$. For each node $i$,

$$h_i = \text{MPNN}(i, \bar{A}, X). \tag{16}$$

For all target links, MPNN needs to run only once.

**Predictor.**  Predictors use the node representations and graph structure to produce link prediction. Link representations of NCN are as follows,

$$z_{ij} = (h_i \odot h_j \| \sum_{\substack{u \in N(i, \bar{A}) \cap \\ N(j, \bar{A})}} h_u), \tag{17}$$

where $\|$ means concatenation, $z_{ij}$ is the representation of link $(i,j)$. $z_{ij}$ composed of two components: two nodes' presentation $h_i \odot h_j$ and representations of nodes within the common neighbor set. The former component is often used in link prediction models (Kipf & Welling, 2016; Yun et al., 2021; Chamberlain et al., 2023), while we propose the latter one for the first time. Link representations are then used to produce link existence probability.

$$\hat{A}_{ij} = \text{sigmoid}(\text{MLP}(z_{ij})), \tag{18}$$

Table 5: Total time(s) needed in one run

|  | Cora | Citeseer | Pubmed | Collab | PPA | Citation2 | DDI |
|---|---|---|---|---|---|---|---|
| **NCN** | 8 | 16 | 28 | 320 | 9375 | 7123 | 546 |
| **NCNC** | 15 | 27 | 54 | 730 | 77385 | 5170 | 1785 |

Table 6: Scalability comparison. $h, h, h''$: the complexity of hash function in BUDDY, where are all $\geq d$. $F$: the dimension of node representations. When predicting the $t$ target links, time and space complexity of existing models can be expressed as $O(B + Ct)$ and $O(D + Et)$ respectively.

| Architecture | Method | B | C | D | E |
|---|---|---|---|---|---|
| MPNN only | GAE | $ndF + nF^2$ | $F^2$ | $nF$ | $F$ |
| MPNN-and-SF | Neo-GNN | $ndF + nF^2 + nd^l$ | $d^l + F^2$ | $nF + nd^l$ | $d^l + F$ |
|  | BUDDY | $ndF + nh$ | $h' + F^2$ | $nF + nh''$ | $F + h'$ |
| SF-then-MPNN | SEAL | $0$ | $d^{l'+1}F + d^{l'}F^2$ | $0$ | $d^{l'+1}F$ |
| MPNN-then-SF | NCN | $ndF + nF^2$ | $dF + F^2$ | $nF$ | $dF$ |
|  | NCNC | $ndF + nF^2$ | $d^2F + dF^2$ | $nF$ | $d^2F$ |

where $\hat{A}_{ij}$ is the probability that link $(i, j)$ exists.

NCNC has a similar form. The only difference is that $\sum_{u \in N(i,\bar{A}) \cap N(j,\bar{A})} h_u$ in Equation (17) is replaced with the follow form:

$$\sum_{\substack{u \in N(i,\bar{A}) \cap \\ N(j,\bar{A})}} h_u + \sum_{\substack{u \in N(j,\bar{A}) - \\ N(i,\bar{A})}} \hat{A}_{iu} h_u + \sum_{\substack{u \in N(i,\bar{A}) - \\ N(j,\bar{A})}} \hat{A}_{ju} h_u. \tag{19}$$

where $\hat{A}_{ab}$ is the link existence probability produced by NCNC.

## D  EXPERIMENTAL SETTINGS

**Computing infrastructure.** We leverage Pytorch Geometric (Fey & Lenssen, 2019) and Pytorch (Paszke et al., 2019) for model development. All experiments are conducted on an Nvidia 4090 GPU on a Linux server.

**Baselines.** We directly use the results reported in (Chamberlain et al., 2023).

**Model hyperparameter.** We use optuna (Akiba et al., 2019) to perform random searches. Hyperparameters were selected to maximize validation score. The best hyperparameters selected for each model can be found in our code.

**Training process.** We utilize Adam optimizer to optimize models and set an epoch upper bound 100. All results of our models are provided from runs with 10 random seeds.

**Computation cost** The total time of each main experiment is shown in Table 5. Reproducing all main results takes 280 GPU hours.

## E  TIME AND SPACE COMPLEXITY

Let $t$ denote the number of target links, $n$ denote the number of nodes in the graph, and $d$ denote the maximum node degree. Existing models' time and space complexity can be expressed in $O(B + Ct)$ and $O(D + Et)$ respectively, where $B, C, D, E$ are irrelevant to $t$. $B, C, D, E$ of models are summarized in Table 6. The derivation of the complexity is as follows. As NCN, GAE, and GNN with separated structural features run MPNN on the original graph, they share similar $ndF + nF^2$ in $B$. Specifically, BUDDY (Chamberlain et al., 2023) uses a simplified MPNN with $ndF$ in $B$. Moreover, Neo-GNN needs to precompute high order graph $A^l$, which takes $O(nd^l)$ time and space.

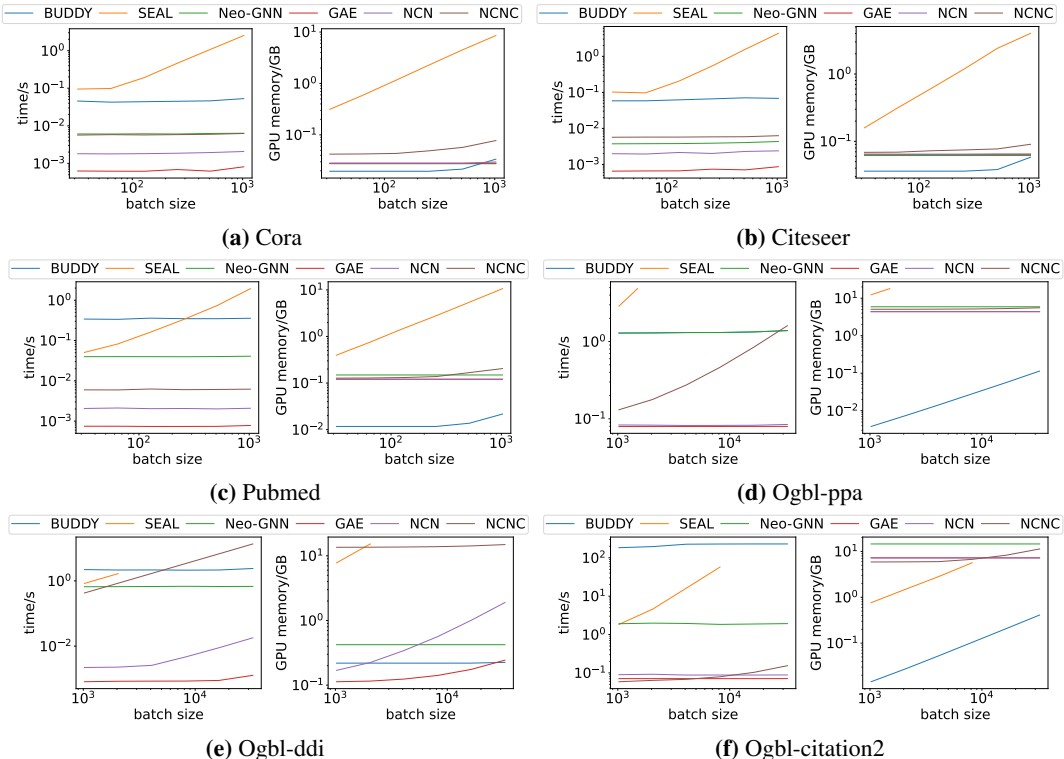

Figure 7: Inference time and GPU memory on datasets. The process we measure includes preprocessing, MPNN, and predicting one batch of test links.

BUDDY needs to hash each node and takes $O(nh)$ time and $O(nh')$ space. In contrast, $B$ of SEAL is 0 as it does not run MPNN on the original graph. For each target link, vanilla GNN only needs to feed the feature vector to MLP for each link, so $C = F^2$. Besides GAE's operation, BUDDY further needs to hash the structure for structural features, whose complexity is complex but higher than $d$ per edge, and Neo-GNN computes pairwise feature with $O(d^l)$ complexity, where $l$ is the number of hop Neo-GNN consider. NCN needs to compute common neighbor: $O(d)$, pool node embeddings: $O(dF)$, and feed to MLP: $O(F^2)$. NCNC-1 runs NCN for each potential common neighbor: $O(F^2 + d(dF + F^2)) = O(d^2F + dF^2)$. Similarly, NCNC-$K$ runs $O(d)$ times NCNC-$(K-1)$, so its time complexity is $O(d^{K+1}F + d^KF^2)$. For each target link, SEAL segregates a subgraph of size $O(d^{l'})$ and runs MPNN on it, so $C = d^{l'}F^2 + d^{l'+1}F$, where $l'$ is the number of hops of the subgraph.

## F    Scalability Comparison on datasets

The time and memory consumption of models on different datasets are shown in Figure 7. On these datasets, we observe results similar to those on the ogbl-collab dataset in Section 6.2: NCN achieves similar computation overhead to GAE; NCNC usually scales better than Neo-GNN; SEAL's scalabilty is the worst. However, on the ogbl-citation2 dataset, SEAL has the lowest GPU memory consumption with small batch sizes, because the whole graph in ogbl-citation2 is large, on which MPNN is expensive, while SEAL only runs MPNN on small subgraphs sampled from the whole graph, leading to lower overhead.

## G    Comparison with other link prediction models

**Node embedding methods**    The main advantage of GNN methods is that they keep permutation equivariance. In other words, these methods can give isomorphic links (links with the same structure) the same prediction. In contrast, node embedding methods, such as Node2Vec (Grover &

Table 7: Results on link prediction benchmarks. The format is average score ± standard deviation. NCN+tricks means NCN with tricks of PLNLP.

|  | **Collab** | **PPA** | **Citation2** | **DDI** |
|---|---|---|---|---|
| Metric | Hits@50 | Hits@100 | MRR | Hits@20 |
| **NCN** | $64.76 \pm 0.87$ | $61.19 \pm 0.85$ | $88.64 \pm 0.14$ | $82.32 \pm 6.10$ |
| **NCNC** | $66.61 \pm 0.71$ | $61.42 \pm 0.73$ | $89.12 \pm 0.40$ | $84.11 \pm 3.67$ |
| **Node2Vec** | $41.36 \pm 0.69$ | $27.83 \pm 2.02$ | $53.47 \pm 0.12$ | $21.95 \pm 1.58$ |
| **DeepWalk** | $50.37 \pm 0.34$ | $28.88 \pm 1.53$ | $84.48 \pm 0.30$ | $26.42 \pm 6.10$ |
| **LINE** | $55.13 \pm 1.35$ | $26.03 \pm 2.55$ | $82.33 \pm 0.52$ | $10.15 \pm 1.69$ |
| **PLNLP** | $70.59 \pm 0.29$ | $32.38 \pm 2.58$ | $84.92 \pm 0.29$ | $90.88 \pm 3.13$ |
| **GIDN** | $70.96 \pm 0.55$ | - | - | - |
| **NCN+tricks** | $68.04 \pm 0.42$ | - | - | $90.83 \pm 2.83$ |

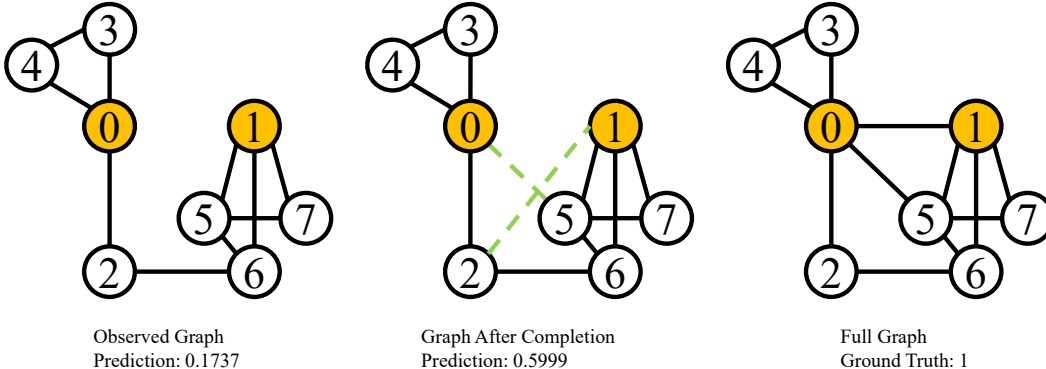

Observed Graph
Prediction: 0.1737

Graph After Completion
Prediction: 0.5999

Full Graph
Ground Truth: 1

Figure 8: Visualization of how NCNC works. The example is from the Cora dataset, a citation graph. The target link is (0, 1). Models should produce high link existence probability. However, in the observed graph, link (0, 5) is missing, and (0,1) thus has no common neighbor. So NCN predicts that the link is not likely to exist. However, for NCNC, it first completes common neighbors (see green lines). Therefore, NCNC predicts that (0, 1) is more likely to exist. Note that NCNC completes common neighbors by probability, and we only plot completion with probability ¿ 0.5 here. And the two completions are with about 0.95 probability. Though the common neighbor 2 completed by the model does not exist in the full graph, the full graph here only means a graph with all training, validation, and test edges, and the citation relation in the graph may still need to be completed.

Leskovec, 2016), LINE (Tang et al., 2015), and DeepWalk (Perozzi et al., 2014), will produce different results for isomorphic links, leading to potentially bad generalization.

We also compare our method with representative node embedding methods on ogb datasets in Table 7. NCN and NCNC outperform node embedding methods significantly on all datasets, indicating the advantages of MPNNs considering pairwise features for link prediction.

**Other GNNs** Instead of representations of pairwise relations, PLNLP (Wang et al., 2021) and GIDN (Wang et al., 2022) boost GNNs on link prediction tasks by training tricks like loss function and data augmentation. These tricks are orthogonal to our model design. In experiments (Table 7), compared with PLNLP, NCN achieves $89\%$ performance gain on ogbl-ppa and $20\%$ gain on average. As GIDN only conducts experiment on one dataset ogbl-collab, the comparison is not complete. Moreover, tricks of PLNLP can also boost our models.

# H CNC EXAMPLE

Figure 8 provides an example from Cora dataset on how CNC works.

## I ABLATION OF MPNN

Here we provide an ablation study on the MPNN used in NCN. The results are shown in Table 8. The MPNN model includes GIN (Xu et al., 2019), GraphSage (Hamilton et al., 2017), MPNN with max aggregation, GCN (Kipf & Welling, 2017), and GAT (Velickovic et al., 2018). Though the performance of NCN is sensitive to the MPNN model, NCN achieves performance gain with all GNNs compared with GraphAutoencoder (GAE).

Table 8: Ablation study on MPNN.

| Dataset | Model | GIN | GraphSage | max | GCN | GAT |
|---------|-------|-----|-----------|-----|-----|-----|
| Cora | GAE | $70.45_{\pm 1.88}$ | $70.59_{\pm 1.70}$ | $61.63_{\pm 4.43}$ | $89.01_{\pm 1.32}$ | $83.36_{\pm 2.54}$ |
|  | NCN | $70.62_{\pm 1.68}$ | $70.94_{\pm 1.47}$ | $66.53_{\pm 2.27}$ | $89.05_{\pm 0.96}$ | $83.93_{\pm 2.03}$ |
| Citeseer | GAE | $61.21_{\pm 1.18}$ | $61.23_{\pm 1.28}$ | $53.02_{\pm 3.75}$ | $91.78_{\pm 0.94}$ | $68.49_{\pm 2.75}$ |
|  | NCN | $61.58_{\pm 1.18}$ | $61.95_{\pm 1.05}$ | $53.40_{\pm 2.34}$ | $91.56_{\pm 1.43}$ | $69.27_{\pm 2.08}$ |
| Pubmed | GAE | $59.00_{\pm 0.31}$ | $57.20_{\pm 1.37}$ | $55.08_{\pm 1.43}$ | $78.81_{\pm 1.64}$ | $74.44_{\pm 1.04}$ |
|  | NCN | $59.06_{\pm 0.49}$ | $58.06_{\pm 0.69}$ | $56.32_{\pm 0.77}$ | $79.05_{\pm 1.16}$ | $74.43_{\pm 0.81}$ |
| collab | GAE | $38.94_{\pm 0.81}$ | $28.11_{\pm 0.26}$ | $27.08_{\pm 0.61}$ | $36.96_{\pm 0.95}$ | OOM |
|  | NCN | $64.38_{\pm 0.06}$ | $63.94_{\pm 0.43}$ | $64.19_{\pm 0.18}$ | $64.76_{\pm 0.87}$ | OOM |
| ppa | GAE | $18.20_{\pm 0.45}$ | $11.79_{\pm 1.02}$ | $20.86_{\pm 0.81}$ | $19.49_{\pm 0.75}$ | OOM |
|  | NCN | $47.94_{\pm 0.89}$ | $56.41_{\pm 0.65}$ | $57.31_{\pm 0.30}$ | $61.19_{\pm 0.85}$ | OOM |

## J CHOICE OF METRICS

We test our model in different metrics. The results are shown in Table 9. In total, NCN achieves 11 best score (in bold), NCNC achieves 22 best score, and our strongest baseline achieves 9 best score. Therefore, our NCN and NCNC still outperforms baselines in different metrics.

Table 9: Models' performance with various metrics. BUDDY is our strongest baseline. Blanks mean unfinished experiments due to time constraints.

|  |  | Cora | Citeseer | Pubmed | Collab | PPA | Citation2 | DDI |
|---|---|------|----------|--------|--------|-----|-----------|-----|
| hit@1 | NCN | $\mathbf{16.24_{\pm 14.18}}$ | $29.32_{\pm 18.19}$ | $7.03_{\pm 6.10}$ | $4.94_{\pm 2.95}$ | $5.91_{\pm 4.11}$ | $83.79_{\pm 0.06}$ | $0.24_{\pm 0.11}$ |
|  | NCNC | $10.90_{\pm 11.40}$ | $\mathbf{32.45_{\pm 17.01}}$ | $\mathbf{8.57_{\pm 6.76}}$ | $9.82_{\pm 2.49}$ | $\mathbf{7.78_{\pm 0.63}}$ |  | $0.16_{\pm 0.07}$ |
|  | BUDDY | $11.74_{\pm 5.77}$ | $20.87_{\pm 12.22}$ | $2.97_{\pm 2.02}$ | $\mathbf{10.71_{\pm 0.64}}$ | $2.29_{\pm 1.26}$ |  | $\mathbf{2.40_{\pm 4.81}}$ |
| hit@3 | NCN | $29.52_{\pm 13.79}$ | $49.98_{\pm 14.49}$ | $\mathbf{19.16_{\pm 4.39}}$ | $11.07_{\pm 6.32}$ | $15.32_{\pm 3.31}$ | $92.41_{\pm 0.06}$ | $1.54_{\pm 3.43}$ |
|  | NCNC | $25.04_{\pm 11.40}$ | $\mathbf{50.49_{\pm 12.01}}$ | $17.58_{\pm 6.57}$ | $\mathbf{21.07_{\pm 5.46}}$ | $\mathbf{16.58_{\pm 0.60}}$ |  | $0.59_{\pm 0.42}$ |
|  | BUDDY | $\mathbf{32.67_{\pm 10.10}}$ | $41.16_{\pm 9.12}$ | $10.41_{\pm 4.16}$ | $16.25_{\pm 1.59}$ | $7.75_{\pm 0.48}$ |  | $\mathbf{10.84_{\pm 7.55}}$ |
| hit@10 | NCN | $\mathbf{55.87_{\pm 4.40}}$ | $69.68_{\pm 3.05}$ | $\mathbf{34.61_{\pm 5.02}}$ | $43.51_{\pm 1.84}$ | $25.76_{\pm 3.65}$ | $96.50_{\pm 0.06}$ | $40.04_{\pm 19.59}$ |
|  | NCNC | $53.78_{\pm 7.33}$ | $69.59_{\pm 4.48}$ | $34.29_{\pm 4.43}$ | $43.22_{\pm 6.19}$ | $\mathbf{26.67_{\pm 1.51}}$ |  | $45.64_{\pm 14.12}$ |
|  | BUDDY | $50.98_{\pm 3.46}$ | $67.05_{\pm 2.83}$ | $23.92_{\pm 5.01}$ | $\mathbf{53.11_{\pm 0.86}}$ | $17.41_{\pm 0.06}$ |  | $\mathbf{52.70_{\pm 7.70}}$ |
| hit@20 | NCN | $\mathbf{68.31_{\pm 3.00}}$ | $78.02_{\pm 1.99}$ | $50.94_{\pm 3.11}$ | $55.87_{\pm 0.36}$ | $\mathbf{37.57_{\pm 1.98}}$ | $97.87_{\pm 0.04}$ | $82.55_{\pm 4.08}$ |
|  | NCNC | $67.10_{\pm 2.96}$ | $\mathbf{79.05_{\pm 2.68}}$ | $\mathbf{51.42_{\pm 3.81}}$ | $57.83_{\pm 3.14}$ | $35.00_{\pm 2.22}$ |  | $\mathbf{83.92_{\pm 3.25}}$ |
|  | BUDDY | $61.92_{\pm 2.67}$ | $76.15_{\pm 3.31}$ | $34.75_{\pm 5.12}$ | $\mathbf{59.06_{\pm 0.57}}$ | $27.28_{\pm 0.52}$ |  | $78.14_{\pm 4.23}$ |
| hit@50 | NCN | $80.85_{\pm 1.12}$ | $86.33_{\pm 1.55}$ | $67.77_{\pm 1.91}$ | $64.45_{\pm 0.35}$ | $\mathbf{51.54_{\pm 1.48}}$ | $99.01_{\pm 0.02}$ | $94.17_{\pm 0.36}$ |
|  | NCNC | $\mathbf{81.36_{\pm 1.86}}$ | $\mathbf{88.60_{\pm 1.51}}$ | $\mathbf{69.25_{\pm 2.87}}$ | $\mathbf{66.88_{\pm 0.66}}$ | $48.66_{\pm 0.18}$ |  | $\mathbf{94.85_{\pm 0.56}}$ |
|  | BUDDY | $76.64_{\pm 2.45}$ | $85.46_{\pm 2.17}$ | $55.75_{\pm 3.38}$ | $66.09_{\pm 0.48}$ | $39.99_{\pm 0.02}$ |  | $92.17_{\pm 0.95}$ |
| hit@100 | NCN | $\mathbf{89.14_{\pm 1.04}}$ | $91.82_{\pm 1.14}$ | $79.56_{\pm 1.11}$ | $67.25_{\pm 0.15}$ | $61.25_{\pm 0.61}$ | $99.51_{\pm 0.02}$ | $97.09_{\pm 0.43}$ |
|  | NCNC | $89.05_{\pm 1.24}$ | $\mathbf{93.13_{\pm 1.13}}$ | $\mathbf{81.18_{\pm 1.24}}$ | $\mathbf{71.96_{\pm 0.14}}$ | $\mathbf{62.02_{\pm 0.74}}$ |  | $\mathbf{97.60_{\pm 0.22}}$ |
|  | BUDDY | $84.82_{\pm 1.96}$ | $91.48_{\pm 1.15}$ | $70.92_{\pm 2.08}$ | $70.53_{\pm 0.17}$ | $48.07_{\pm 0.05}$ |  | $95.38_{\pm 0.65}$ |
| mrr | NCN | $\mathbf{29.20_{\pm 13.59}}$ | $43.93_{\pm 12.87}$ | $\mathbf{17.44_{\pm 3.40}}$ | $13.76_{\pm 2.49}$ | $13.48_{\pm 2.83}$ | $88.62_{\pm 0.05}$ | $5.48_{\pm 1.23}$ |
|  | NCNC | $23.55_{\pm 9.67}$ | $\mathbf{45.64_{\pm 11.78}}$ | $15.63_{\pm 4.13}$ | $17.68_{\pm 2.70}$ | $\mathbf{14.37_{\pm 0.06}}$ |  | $8.61_{\pm 1.37}$ |
|  | BUDDY | $27.28_{\pm 4.71}$ | $35.77_{\pm 9.59}$ | $10.79_{\pm 2.81}$ | $\mathbf{18.97_{\pm 0.50}}$ | $7.47_{\pm 0.02}$ |  | $\mathbf{13.53_{\pm 6.07}}$ |

