# OpenReview forum: "Neural Common Neighbor with Completion for Link Prediction"
_ICLR.cc/2024/Conference — ICLR 2024 poster_

### Official Review · Reviewer_nThn · 2023-11-01

**Soundness:** 2 fair
**Presentation:** 2 fair
**Contribution:** 2 fair
**Rating:** 3
**Confidence:** 4

**Summary:**

In this paper, the authors propose a novel link framework, which they combine the MPNN and the SF. It is interesting to investigate the combination of the MPNN and the SF. However, the authors mainly compare their method against solely using the MPNN and solely using the SF. Another issue lies in the time complexity analysis. It would much better to report the time complexity of their paper compared to solely using the MPNN and the SF and the other combination methods.

**Strengths:**

1. It is interesting to study the combination of the MPNN and the SF.
2. It is great to see the improvement of the proposed method against the baselines.
3. Summarizing the limitations is always encouraged.

**Weaknesses:**

1. The theoretical analysis is mainly on the comparison between the combinations of the MPNN and the SF against solely using the MPNN and the SF.
2. There is no complexity analysis.
3. There is no clear comparisons among different combinations (as shown in Figure 2).

**Questions:**

Firstly, in the introduction section, the authors have described three ways to combine the MPNN and the SF. However, there are no clear comparisons among these approaches. For example, what use cases fit each approach, and what is the complexity cost of each way? Second, the description of their approach is in Section 3.2, which is very simple and straightforward. I expect more analysis on why the proposed method is better than other combinations, instead of proving why the proposed method is better than solely using the MPNN and solely using the SF. For the experiment, I consider the proposed framework to be a general framework that can be applied to various GNNs. Therefore, I expect to see the results of employing the proposed method in different GNNs to verify the generalizability of the proposed method. Moreover, I also want to ask the authors why to report the results in terms of different evaluation metrics in Table 3. Could you please provide a completed version in the appendix? Also, in Table 3, we can see in some datasets such as Core, CIteseer, and Pubmed, there is very slight difference between GAE and NCN. I highly recommend the authors to further investigate the best use cases of the proposed method. Furthermore, it would be very interesting to summarize a table that includes the pros, the cons, and the best use cases, the worse use cases of different combinations in Figure 2. In Section 6.2, the authors report the memory costs, I highly recommend the authors to report the time complexity cost of applying the proposed methods against solely applying the MPNN and solely applying the SF, as the time complexity is quite crucial to determine whether a model can be applied online or not.
Overall, I do not think this paper is ready for publication for this version. Please correct me if I have some misunderstanding.

---

> ### Author Response · Authors · 2023-11-16
> **Response to Reviewer nThn (1/4)**
>
> We acknowledge and appreciate Reviewer tCCB's detailed review. In addressing the concerns raised, we hope to clarify any misunderstandings and provide further insight into our work.
>
> 1. **W1:** The theoretical analysis is mainly on the comparison between the combinations of the MPNN and the SF against solely using the MPNN and the SF.
>
> We would like to first clarify that our Theorem 1 proves that our MPNN-then-SF architecture (illustrated in Figure 2) is strictly more expressive than GAE, CN, RA, AA, Neo-GNN, and BUDDY. Among them, Neo-GNN and BUDDY are exactly MPNN-and-SF methods. Thus, we are not solely showing our MPNN-and-SF is better than MPNN alone or SF alone, but also showing MPNN-then-SF is better than MPNN-and-SF.
>
> One future work is to theoretically compare MPNN-then-SF and SF-then-MPNN. It is recognized in existing literature [1] that the SF-then-MPNN architecture can achieve maximal expressivity under certain conditions. Whether MPNN-then-SF can also reach maximal expressivity remains unknown. We will continue working on it.
>
>
> [1] Zhang M, Chen Y. Link prediction based on graph neural networks. NeurIPS, 2018.
>
> 2. **W2:** There is no complexity analysis.
>
> We would like to point out that we have included theoretical time and space complexity details in Appendix E and Table 6 in our submission, as referenced at the end of section 4. Our method demonstrates comparable or lower complexity than existing models. Empirical scalability, as shown in Figure 5 and detailed in Appendix F, indicates that our NCN is as efficient as GAE in terms of time and space and more scalable compared to other performative GNNs for link prediction.
>
> 3. **W3, Q1 and Q6:** There is no clear comparisons among different combinations as shown in Figure 2 (W3). In the introduction section, the authors have described three ways to combine the MPNN and the SF. However, there are no clear comparisons among these approaches. For example, what use cases fit each approach, and what is the complexity cost of each way (Q1)? Furthermore, it would be very interesting to summarize a table that includes the pros, the cons, and the best use cases, the worse use cases of different combinations in Figure 2 (Q6).
>
>
> We have included a textual comparison among different combinations in Section 4.1 of our submission. But we agree with the reviewer that a table of summary can make our paper more clear. For clarity, we summarize the pros and cons of each combination into a table:
>
> | Architecture        | Overview                                                    | theoretical expressivity | scalability | Empirical Performance |
> | ------------------- | ----------------------------------------------------------- | ------------------------ | ----------- | -------------------------- |
> | SF only             | manual structural features                                  | poor                     | good        | poor                       |
> | GAE (MPNN only)     | Two  target node's MPNN representation                      | poor                     | good        | poor                       |
> | SF-and-MPNN         | Concatenating SF and two  target node's MPNN representation | poor                     | good        | good                       |
> | SF-then-MPNN        | MPNN takes graph augmented by SF as input.                  | good                     | poor        | good                       |
> | MPNN-then-SF (ours) | Use SF to guide the pooling of MPNN node representation.    | good                     | good        | best                     |
>
>
> Our MPNN-then-SF architecture consistently outperforms existing models in real-world datasets, achieving both high scalability and expressivity. Their time complexity varies with the detailed implementation. However, our MPNN-then-SF achieves the lowest complexity among existing models with other architectures, as shown in our response to W2.
>
> The MPNN-and-SF combination is particularly suited for scenarios where the dataset is large, and scalability is a critical concern. On the other hand, the SF-then-MPNN approach is more appropriate for small graphs with complex structures where high expressivity is essential. Our MPNN-then-SF model balances scalability and expressivity, making it a versatile solution applicable to both scenarios.

---

> ### Author Response · Authors · 2023-11-16
> **Response to Reviewer nThn (2/4)**
>
> 5. **Q2:** Second, the description of their approach is in Section 3.2, which is very simple and straightforward. I expect more analysis on why the proposed method is better than other combinations, instead of proving why the proposed method is better than solely using the MPNN and solely using the SF.
>
> As shown in our response to W1, our Theorem 1 in the paper has proved that our model is more expressive than models combining MPNN and SF with the MPNN-and-SF architecture. The expressivity comparison between our MPNN-then-SF and the other previous architecture, SF-then-MPNN, remains unclear. The advantage of our MPNN-then-SF over SF-then-MPNN is the better scalability and empirical performance.
>
> Our analysis in W3 highlights the unique balance of scalability and expressivity in our MPNN-then-SF model, distinguishing it from other combination methods, namely MPNN-and-SF and SF-then-MPNN. The distinct advantages arise from how each model integrates SF and MPNN.
>
> * **Expressivity:** The coupling of SF and MPNN remains crucial for expressivity (see Figure 3 and Theorem 1 in our paper). However, MPNN-and-SF simply concatenates them, leading to lower expressivity than our MPNN-then-SF.
> * **Scalability:** SF changes with the target link. As the MPNN in SF-then-MPNN architecture takes a graph augmented by SF, it has to rerun for different target links. In contrast, our MPNN-then-SF architecture uses the original graph as input and runs MPNN only once for all target links and thus has better scalability.
>
>
>
> 6. **Q3:** For the experiment, I consider the proposed framework to be a general framework that can be applied to various GNNs. Therefore, I expect to see the results of employing the proposed method in different GNNs to verify the generalizability of the proposed method.
>
> We agree with the reviewer that a general framework should be applicable to various GNNs. We have added an ablation study on different GNN models in Appendix I of our updated paper. We also show the results below. The NCN's performance varies with the choice of MPNN model but consistently shows improvements over GAE, demonstrating the method's generalizability.
>
> | Dataset  | Model | GIN                | GraphSage          | max                | GCN                | GAT                |
> |----------|-------|--------------------|--------------------|--------------------|--------------------|--------------------|
> | Cora     | GAE   | $70.45_{\pm 1.88}$ | $70.59_{\pm 1.70}$ | $61.63_{\pm 4.43}$ | $89.01_{\pm 1.32}$ | $83.36_{\pm 2.54}$ |
> |          | NCN   | $70.62_{\pm 1.68}$ | $70.94_{\pm 1.47}$ | $66.53_{\pm 2.27}$ | $89.05_{\pm 0.96}$ | $83.93_{\pm 2.03}$ |
> | Citeseer | GAE   | $61.21_{\pm 1.18}$ | $61.23_{\pm 1.28}$ | $53.02_{\pm 3.75}$ | $91.78_{\pm 0.94}$ | $68.49_{\pm 2.75}$ |
> |          | NCN   | $61.58_{\pm 1.18}$ | $61.95_{\pm 1.05}$ | $53.40_{\pm 2.34}$ | $91.56_{\pm 1.43}$ | $69.27_{\pm 2.08}$ |
> | Pubmed   | GAE   | $59.00_{\pm 0.31}$ | $57.20_{\pm 1.37}$ | $55.08_{\pm 1.43}$ | $78.81_{\pm 1.64}$ | $74.44_{\pm 1.04}$ |
> |          | NCN   | $59.06_{\pm 0.49}$ | $58.06_{\pm 0.69}$ | $56.32_{\pm 0.77}$ | $79.05_{\pm 1.16}$ | $74.43_{\pm 0.81}$ |
> | collab   | GAE   | $38.94_{\pm 0.81}$ | $28.11_{\pm 0.26}$ | $27.08_{\pm 0.61}$ | $36.96_{\pm 0.95}$ | OOM                |
> |          | NCN   | $64.38_{\pm 0.06}$ | $63.94_{\pm 0.43}$ | $64.19_{\pm 0.18}$ | $64.76_{\pm 0.87}$ | OOM                |
> | ppa      | GAE   | $18.20_{\pm 0.45}$ | $11.79_{\pm 1.02}$ | $20.86_{\pm 0.81}$ | $19.49_{\pm 0.75}$ | OOM                |
> |          | NCN   | $47.94_{\pm 0.89}$ | $56.41_{\pm 0.65}$ | $57.31_{\pm 0.30}$ | $61.19_{\pm 0.85}$ | OOM                |

---

> ### Author Response · Authors · 2023-11-16
> **Response to Reviewer nThn (3/4)**
>
> 7. **Q4:** Moreover, I also want to ask the authors why to report the results in terms of different evaluation metrics in Table 3. Could you please provide a completed version in the appendix?
>
> We chose metrics in line with the Open Graph Benchmark [1] standard and previous representative studies [2]. For completion, a more comprehensive evaluation using additional metrics (Hits@1, Hits@3, Hits@10, Hits@20, Hits@50, Hits@100, MRR) is added in Appendix J in the revision. The results are also shown as follows.
>
> |  Metrics| Model| Cora| Citeseer| Pubmed | Collab | PPA |  DDI |
> |---------|-------|--------|------|-------------|-------------|------------|------------|
> | hit@1| NCN| $\mathbf{16.24_{\pm 14.18}}$ | $29.32_{\pm 18.19}$ | $7.03_{\pm 6.10}$| $4.94_{\pm 2.95}$| $5.91_{\pm 4.11}$| $0.24_{\pm 0.11}$|
> |  | NCNC  | $10.90_{\pm 11.40}$ | $\mathbf{32.45_{\pm 17.01}}$ | $\mathbf{8.57_{\pm 6.76}}$  | $9.82_{\pm 2.49}$| $\mathbf{7.78_{\pm 0.63}}$  | $0.16_{\pm 0.07}$|
> |  | BUDDY | $11.74_{\pm 5.77}$| $20.87_{\pm 12.22}$ | $2.97_{\pm 2.02}$| $\mathbf{10.71_{\pm 0.64}}$ | $2.29_{\pm 1.26}$| $\mathbf{2.40_{\pm 4.81}}$  |
> | hit@3| NCN| $29.52_{\pm 13.79}$ | $49.98_{\pm 14.49}$ | $\mathbf{19.16_{\pm 4.39}}$ | $11.07_{\pm 6.32}$ | $15.32_{\pm 3.31}$ | $1.54_{\pm 3.43}$|
> |  | NCNC  | $25.04_{\pm 11.40}$ | $\mathbf{50.49_{\pm 12.01}}$ | $17.58_{\pm 6.57}$ | $\mathbf{21.07_{\pm 5.46}}$ | $\mathbf{16.58_{\pm 0.60}}$ | $0.59_{\pm 0.42}$|
> |  | BUDDY | $\mathbf{32.67_{\pm 10.10}}$ | $41.16_{\pm 9.12}$| $10.41_{\pm 4.16}$ | $16.25_{\pm 1.59}$ | $7.75_{\pm 0.48}$| $\mathbf{10.84_{\pm 7.55}}$ |
> | hit@10  | NCN| $\mathbf{55.87_{\pm 4.40}}$  | $\mathbf{69.68_{\pm 3.05}}$  | $\mathbf{34.61_{\pm 5.02}}$ | $43.51_{\pm 1.84}$ | $25.76_{\pm 3.65}$ | $40.04_{\pm 19.59}$|
> |  | NCNC  | $53.78_{\pm 7.33}$| $69.59_{\pm 4.48}$| $34.29_{\pm 4.43}$ | $43.22_{\pm 6.19}$ | $\mathbf{26.67_{\pm 1.51}}$ | $45.64_{\pm 14.12}$|
> |  | BUDDY | $50.98_{\pm 3.46}$| $67.05_{\pm 2.83}$| $23.92_{\pm 5.01}$ | $\mathbf{53.11_{\pm 0.86}}$ | $17.41_{\pm 0.06}$ | $\mathbf{52.70_{\pm 7.70}}$ |
> | hit@20  | NCN| $\mathbf{68.31_{\pm 3.00}}$  | $78.02_{\pm 1.99}$| $50.94_{\pm 3.11}$ | $55.87_{\pm 0.36}$ | $\mathbf{37.57_{\pm 1.98}}$ | $82.55_{\pm 4.08}$ |
> |  | NCNC  | $67.10_{\pm 2.96}$| $\mathbf{79.05_{\pm 2.68}}$  | $\mathbf{51.42_{\pm 3.81}}$ | $57.83_{\pm 3.14}$ | $35.00_{\pm 2.22}$ | $\mathbf{83.92_{\pm 3.25}}$ |
> |  | BUDDY | $61.92_{\pm 2.67}$| $76.15_{\pm 3.31}$| $34.75_{\pm 5.12}$ | $\mathbf{59.06_{\pm 0.57}}$ | $27.28_{\pm 0.52}$ | $78.14_{\pm 4.23}$ |
> | hit@50  | NCN| $80.85_{\pm 1.12}$| $86.33_{\pm 1.55}$| $67.77_{\pm 1.91}$ | $64.45_{\pm 0.35}$ | $\mathbf{51.54_{\pm 1.48}}$  | $94.17_{\pm 0.36}$ |
> |  | NCNC  | $\mathbf{81.36_{\pm 1.86}}$  | $\mathbf{88.60_{\pm 1.51}}$  | $\mathbf{69.25_{\pm 2.87}}$ | $\mathbf{66.88_{\pm 0.66}}$ | $48.66_{\pm 0.18}$| $\mathbf{94.85_{\pm 0.56}}$ |
> |  | BUDDY | $76.64_{\pm 2.45}$| $85.46_{\pm 2.17}$| $55.75_{\pm 3.38}$ | $66.09_{\pm 0.48}$ | $39.99_{\pm 0.02}$ | $92.17_{\pm 0.95}$ |
> | hit@100 | NCN| $\mathbf{89.14_{\pm 1.04}}$  | $91.82_{\pm 1.14}$| $79.56_{\pm 1.11}$ | $67.25_{\pm 0.15}$ | $61.25_{\pm 0.61}$ | $97.09_{\pm 0.43}$ |
> |  | NCNC  | $89.05_{\pm 1.24}$| $\mathbf{93.13_{\pm 1.13}}$  | $\mathbf{81.18_{\pm 1.24}}$ | $\mathbf{71.96_{\pm 0.14}}$ | $\mathbf{62.02_{\pm 0.74}}$ | $\mathbf{97.60_{\pm 0.22}}$ |
> |  | BUDDY | $84.82_{\pm 1.96}$| $91.48_{\pm 1.15}$| $70.92_{\pm 2.08}$ | $70.53_{\pm 0.17}$ | $48.07_{\pm 0.05}$ | $95.38_{\pm 0.65}$ |
> | mrr| NCN| $\mathbf{29.20_{\pm 13.59}}$ | $43.93_{\pm 12.87}$ | $\mathbf{17.44_{\pm 3.40}}$ | $13.76_{\pm 2.49}$ | $13.48_{\pm 2.83}$ | $5.48_{\pm 1.23}$|
> |  | NCNC  | $23.55_{\pm 9.67}$| $\mathbf{45.64_{\pm 11.78}}$ | $15.63_{\pm 4.13}$ | $17.68_{\pm 2.70}$ | $\mathbf{14.37_{\pm 0.06}}$ | $8.61_{\pm 1.37}$|
> |  | BUDDY | $27.28_{\pm 4.71}$| $35.77_{\pm 9.59}$| $10.79_{\pm 2.81}$ | $\mathbf{18.97_{\pm 0.50}}$ | $7.47_{\pm 0.02}$| $\mathbf{13.53_{\pm 6.07}}$ |
>
> In total, NCN achieves best score in 11 cases (in bold), NCNC achieves 22 best score, and our strongest baseline BUDDY achieves 9 best score. Therefore, our NCN and NCNC still outperforms baselines in different metrics.
>
> [1] Hu W, et al. Open Graph Benchmark: Datasets for Machine Learning on Graphs. NeurIPS 2020.
> [2] Chamberlain B, et al. Graph neural networks for link prediction with subgraph sketching. ICML 2023.

---

> ### Author Response · Authors · 2023-11-16
> **Response to Reviewer nThn (4/4)**
>
> 8. **Q5:** Also, in Table 3, we can see in some datasets such as Cora, CIteseer, and Pubmed, there is very slight difference between GAE and NCN. I highly recommend the authors to further investigate the best use cases of the proposed method.
>
>
> We thank the reviewer for insightful question. In the citation datasets (like Cora, Pubmed, and Citeseer), node features (the contents of the articles) are more important for determining whether one article cites another (form a link), while structural features are less significant (refer to Table 3, where the SF-only method CN performs poorly on these datasets). Therefore, the combination of MPNN and SF does not yield a significant performance gain in these cases. However, when the structural features are important, or the node feature is not very informative, our NCN achieves a much larger performance gain than Graph Autoencoder (GAE). For instance, we conducted an experiment where we disregarded the node features of the datasets, using all zeros instead. The results are as follows:
>
> |      | Cora       | Citeseer   | Pubmed     |
> | ---- | ---------- | ---------- | ---------- |
> | CN   | 33.92±0.46 | 29.79±0.90 | 23.13±0.15 |
> | GAE  | 36.07±1.45 | 31.50±2.14 | 14.61±1.10 |
> | NCN  | 60.49±1.01 | 51.76±2.40 | 35.22±0.89 |
>
> In this table, NCN significantly outperforms both CN and GAE, validating the effectiveness of our MPNN-then-SF architecture.
>
> Moreover, even in scenarios where the contribution of SF is less pronounced, NCN still maintains comparable or superior performance to GAE with similar costs. This makes NCN a universal choice for **most cases**.
>
>
>
> 9. **Q7:**  In Section 6.2, the authors report the memory costs, I highly recommend the authors to report the time complexity cost of applying the proposed methods against solely applying the MPNN and solely applying the SF, as the time complexity is quite crucial to determine whether a model can be applied online or not.
>
> We have reported the time costs in the first subfigure of Figure 5 in Section 6.2 in our submission. In this subfigure, GAE represents the method utilizing MPNN only. Additionally, for a comprehensive scalability comparison, we have included CN (which applies solely SF) in the revised version. Our NCN demonstrates high scalability, achieving a time cost comparable to that of GAE.

---

> > ### Author Response · Authors · 2023-11-22
> > **Discussion Stage Closing Soon**
> >
> > Dear Reviewer nThn,
> >
> > We would like to express our gratitude for your valuable feedback and constructive comments on our paper. We have carefully considered and addressed all of your comments and suggestions in our response. However, we have not received any further communication from you. Given the importance of your feedback to the fair evaluation of our paper, we kindly request your prompt response.
> >
> > To summarize our responses:
> >
> > W1: We clarified the theoretical analysis and highlighted the superiority of our MPNN-then-SF architecture over MPNN-and-SF rather than MPNN only and SF only.
> >
> > W2: We provided details on the time and space complexity. Our NCN achieves comparable or lower complexity than other models.
> >
> > W3, Q1, and Q6: We added a clear comparison among different combinations in a table, highlighting the pros and cons of each architecture. This table provides a comprehensive overview of the strengths and weaknesses of various combinations.
> >
> > Q2: We highlight the unique advantages of our MPNN-then-SF model in terms of scalability and expressivity compared with other combination of MPNN and SF.
> >
> > Q3: We added an ablation study in Appendix I to demonstrate the generalizability of our method when applied to different GNN models.
> >
> > Q4: We included additional evaluation metrics in Appendix J for a more comprehensive analysis of our model's performance.
> >
> > Q7: We have reported the time complexity costs in Figure 5 and have included CN for further scalability comparison.
> >
> >
> > Your prompt response would greatly assist us in proceeding with the revisions.
> >
> > Thank you once again for your time and effort in reviewing our work. We look forward to hearing from you soon.
> >
> > Best, Authors`

---

> > > ### Author Response · Authors · 2023-11-23
> > > **Last Day of Discussion Stage**
> > >
> > > Dear Reviewer nThn,
> > >
> > > We genuinely appreciate your meticulous review of our paper. We have taken great care to thoroughly address all of your valuable comments and suggestions in our response. However, we have yet to receive any additional communication from you, and with only 3 hours remaining for the discussion phase, your input is crucial to ensure a fair evaluation of our work. We eagerly await your response and hope to hear from you soon.
> > >
> > > Best, Authors`

---

### Official Review · Reviewer_tCCB · 2023-11-02

**Soundness:** 4 excellent
**Presentation:** 3 good
**Contribution:** 3 good
**Rating:** 8
**Confidence:** 4

**Summary:**

In this paper, the authors propose a novel link prediction model called NCN. Unlike prior works, which can be classified into SF-then-MPNN and SF-and-MPNN categories, NCN introduces a novel architecture, MPNN-then-SF, that overcomes the defects of the previous two methods and offers a unique combination of high expressivity and scalability. Further addressing the Common Neighbor (CN) distribution shift problem caused by the graph incompleteness in existing link prediction settings, the authors introduce the Common Neighbor Structure Completion module to enhance NCN's performance. Finally, through extensive experiments on seven commonly used link prediction datasets, the author demonstrates the model's effectiveness and establishes it as the new state-of-the-art model for link prediction tasks.

**Strengths:**

- The paper is well-written, and the idea is easy to follow. For most claims, the paper provides either theoretical proofs or empirical results to support them, demonstrating its soundness.

- Extensive experiments have been conducted on commonly used link prediction benchmark datasets, with most baselines being competitive. The paper also includes details about efficiency statistics (time and resource consumption) and parameter settings.

- The observation regarding the CN distribution shift is very interesting and could inspire future work aimed at improving the performance of link prediction models by addressing this issue.

**Weaknesses:**

There are some minor issues with notation. For example, in Equation 6, why are double brackets used, "{{" and "}}"? In Equation 9, it would be better to explain the symbol "||" as concatenation again, as the definition of "||" is provided at the end of section 3.1, which is quite far away and might cause confusion for readers.

**Questions:**

How does the method behave on graphs without node features?

---

> ### Author Response · Authors · 2023-11-16
> **Response to Reviewer tCCB**
>
> We are grateful for Reviewer tCCB's thorough review and address the points raised as follows:
>
> * **W1:** There are some minor issues with notation. For example, in Equation 6, why are double brackets used, "{{" and "}}"? In Equation 9, it would be better to explain the symbol "|\|" as concatenation again, as the definition of "|\|" is provided at the end of section 3.1, which is quite far away and might cause confusion for readers.
>
> We agree the notation issues pointed out by Reviewer tCCB and have fixed them in revision.
>
> * **Q1:** How does the method behave on graphs without node features?
>
> The performance of NCN on graphs without node features is indeed a valuable aspect to investigate, particularly since one of NCN's strengths lies in integrating node features with structural features. To address this, we conducted experiments on various datasets, intentionally omitting node features. Notably, among these datasets, the DDI dataset inherently lacks node features. The following results were obtained:
>
> |      | Cora       | Citeseer   | Pubmed     | Collab     | PPA        | DDI        |
> | ---- | ---------- | ---------- | ---------- | ---------- | ---------- | ---------- |
> | CN   | 33.92±0.46 | 29.79±0.90 | 23.13±0.15 | 56.44±0.00 | 27.65±0.00 | 17.73±0.00 |
> | GAE  | 36.07±1.45 | 31.50±2.14 | 14.61±1.10 | 0.00±0.00  | 9.27±0.11  | 61.53±9.59 |
> | NCN  | 60.49±1.01 | 51.76±2.40 | 35.22±0.89 | 52.36±0.00 | 57.53±0.75 | 82.32±3.67 |
>
> NCN consistently outperforms the Graph Auto-Encoder (GAE) in scenarios lacking node features. Furthermore, NCN significantly surpasses the performance of Common Neighbors (CN), demonstrating its superior capability in capturing more comprehensive structural features than manual features. These results underscore the robustness of our approach in environments where node features are not available, thereby confirming the effectiveness and necessity of combining SF and MPNN in our proposed framework.

---

> > ### Comment · Reviewer_tCCB · 2023-11-21
> > **Response to author rebuttal**
> >
> > Dear authors, thank you for answering my questions. I have no additional need for any clarifications and I will stand by my original accept decision.

---

### Official Review · Reviewer_Ztrp · 2023-11-04

**Soundness:** 3 good
**Presentation:** 3 good
**Contribution:** 3 good
**Rating:** 6
**Confidence:** 4

**Summary:**

This paper addresses the challenge of GNN-based link prediction, presenting a new framework MPNN-then-SF, aimed at enhancing the expressiveness of structure feature-based link prediction methods. The framework, operationalized through the NCN implementation, incorporates common neighbor features beyond simple counts. This work also tackles the issue of graph incompleteness that can lead to suboptimal link prediction results. The authors propose a common neighbor completion strategy to enrich the input graph. Experimental findings indicate that this methodology outperforms baseline models significantly.

**Strengths:**

1.	The proposed MPNN-then-SF framework innovatively incorporates common neighbor features beyond simple counts, enhancing the expressiveness of the graph-based link prediction model. NCNC further addresses the problem of graph incompleteness through the common neighbor completion module.
2.	The exploration into how graph completeness affects prediction outcomes is insightful. It underscores the importance of a comprehensive input graph for robust link prediction, setting the stage for further research in this area.
3.	The results, as evidenced by Table 2 and Figure 5, show considerable improvements in expressiveness and scalability. The detailed analysis in Table 3 underscores the contribution of each component within the proposed framework, validating its effectiveness.

**Weaknesses:**

1.	While Figure 5 suggests that inference time does not increase significantly, calculating P_{uij} values for all node pairs could, in theory, create a computational burden. The paper would benefit from a more detailed explanation of this process. Is there a mechanism, such as pre-calculation or an efficient online algorithm, that mitigates the computational load during the inference phase?
2.	The paper reveals that a first-order neighborhood is adequate by comparing NCN and NCN2 models. However, the potential of incorporating a broader neighborhood scope remains unclear. Could extending the common neighbor completion (CNC) to consider 2-hop neighbors improve performance significantly? It would be beneficial for the paper to discuss the implications of such an extension and, if possible, to present experimental results from implementing a 2-hop neighbor completion, referred to as NCNC, for a thorough comparison.

**Questions:**

See weaknesses.

---

> ### Author Response · Authors · 2023-11-16
> **Response to Reviewer Ztrp**
>
> We appreciate Reviewer Ztrp's insightful comments and address the concerns as follows:
>
> 1. **W1:** While Figure 5 suggests that inference time does not increase significantly, calculating $P_{uij}$ values for all node pairs could, in theory, create a computational burden. The paper would benefit from a more detailed explanation of this process. Is there a mechanism, such as pre-calculation or an efficient online algorithm, that mitigates the computational load during the inference phase?
>
> We agree that computing $P_{uij}$ for all node pairs create a computational burden. The high scalability is because NCNC does not need to compute for all node pairs. As shown in Equation 10 in our paper, for a given link $(i,j)$, we set $P_{uij}$ to $0$ if $u$ is not a neighbor of either $i$ or $j$. This approach significantly reduces the computational overhead. Specifically, NCN requires $O(d)$  time and space, and NCNC takes $O(d^2)$  time and space to predict a link, where $d$ is the maximal node degree. Our paper shows further details of time and space complexity in Appendix E.
>
> 2. **W2:** The paper reveals that a first-order neighborhood is adequate by comparing NCN and NCN2 models. However, the potential of incorporating a broader neighborhood scope remains unclear. Could extending the common neighbor completion (CNC) to consider 2-hop neighbors improve performance significantly? It would be beneficial for the paper to discuss the implications of such an extension and, if possible, to present experimental results from implementing a 2-hop neighbor completion, referred to as NCNC, for a thorough comparison.
>
> We agree with the reviewer that incorporating a broader neighborhood scope is not fully explored in our paper. However, our objective was not to demonstrate the ineffectiveness of broader neighborhoods. Indeed, prior work [1] has substantiated the utility of higher-order neighbors in SF-and-MPNN architectures. However, in our MPNN-then-SF architecture, incorporating broader neighborhoods offers minimal performance improvement in experiments. This could be attributed to the capability of MPNNs to capture multi-hop neighborhoods for each node, allowing the subsequent SF process, with MPNN node representations as input, to include broader neighborhoods through one-hop neighbors implicitly.
>
> We also experimented with a 2-hop neighbor completion model, NCNC-2hop. This model potentially incurs $O(d^4)$ time per link, where $d$ is the maximal node degree. Due to time constraints, we only evaluated it on the two smallest datasets. The results are as follows:
>
> |           | Citeseer   | Cora       |
> | --------- | ---------- | ---------- |
> | NCN       | 91.56±1.43 | 89.05±0.96 |
> | NCNC      | 93.47±0.95 | 89.65±1.36 |
> | NCN-2hop  | 91.36±1.02 | 88.87±1.34 |
> | NCNC-2hop | 91.63±1.40 | 90.28±1.73 |
>
> While NCNC-2hop demonstrates performance gains compared to NCN-2hop, it underperforms on Citeseer and achieves similar scores to NCNC on Cora. These findings further substantiate that higher-order neighbors do not necessarily contribute significantly to the performance in SF-and-MPNN architectures.
>
> [1] Chamberlain B, et al. Graph neural networks for link prediction with subgraph sketching. ICML, 2023.

---

### Author Response · Authors · 2023-11-19
**Looking Forward to Your Reply**

Dear Reviewers,

Thanks for your time in reviewing our paper. We got many constructive questions and valuable feedback, and have answered these questions in detail. However, we have not received your responses. The discussion stage only has three days left. Could you please take some time to read our rebuttal?

We are looking forward to your further comment on our work.

Best, Authors

---

> ### Author Response · Authors · 2023-11-21
> **Discussion Stage Closing Soon**
>
> Dear Reviewers,
>
> We sincerely appreciate your dedicated time and effort in reviewing our paper. As the discussion stage is rapidly coming to a close, we kindly request your prompt attention to our rebuttal. Your valuable insights and feedback are integral to ensuring a fair and comprehensive evaluation of our paper.
>
> We eagerly anticipate your continued input on our work. If you still have any lingering questions or unresolved concerns, please don't hesitate to reach out. We are more than willing to provide further clarification and address any issues that may arise.
>
> Thank you once again for your invaluable contributions to the review process.
>
> Best, Authors

---

### Meta-Review · Area_Chair_jf6b · 2023-12-11

**Metareview:**

This paper proposes MPNN-then-SF, a novel link prediction model that combines neural common neighbor (NCN) with structural feature (SF) guidance. Additionally, the impact of graph incompleteness is investigated, leading to the introduction of Neural Common Neighbor with Completion (NCNC), which outperforms recent baselines and achieves state-of-the-art results in link prediction benchmarks. Generally, I think this paper is interesting: the proposed MPNN-then-SF and NCNC models both look novel to me. Regarding the only one (of three) negative review, I think the authors' rebuttal comments have well addressed the reviewer's concerns. In sum, I would like to recommend accepting this paper.

**Justification For Why Not Higher Score:**

A negative review with some concerns on the absence of the complexity analysis, and detailed comparison of different combinations.

**Justification For Why Not Lower Score:**

Generally, I think this paper is interesting: the proposed MPNN-then-SF and NCNC models both look novel to me. Regarding the only one (of three) negative review, I think the authors' rebuttal comments have well addressed the reviewer's concerns. In sum, I would like to recommend accepting this paper.

---

### Decision · Program_Chairs · 2024-01-16

Accept (poster)